# PROVABLY CONVERGENT AND PRIVATE DISTRIBUTED OPTIMIZATION VIA SMOOTHED NORMALIZATION

## ABSTRACT

Federated learning enables training machine learning models while preserving the privacy of participants. Surprisingly, there is no differentially private distributed method for smooth, non-convex optimization problems with convergence guarantees. The reason is that standard privacy techniques require bounding the participants' contributions, usually enforced via *clipping* of the updates. Existing literature typically ignores the effect of clipping by assuming the boundedness of gradient norms or analyzes distributed algorithms with clipping, but ignores DP constraints. In this work, we study an alternative approach via *smoothed normalization* of the updates, motivated by its favorable performance in the single-node setting. By integrating smoothed normalization with an Error Compensation mechanism, we design a new distributed algorithm $\alpha$-NormEC. We prove that our method achieves a superior convergence rate over prior works. By extending $\alpha$-NormEC to the DP setting, we obtain the first differentially private distributed optimization algorithm with provable convergence guarantees. Finally, our empirical results from neural network training indicate robust convergence of $\alpha$-NormEC across different parameter settings.

## 1 INTRODUCTION

Federated Learning (FL) has become a viable approach for distributed collaborative training of machine learning models (Konečný et al., 2016; McMahan et al., 2017; 2018). This growing interest has spurred the development of novel distributed optimization methods tailored for FL, focusing on ensuring high *communication efficiency* (Kairouz et al., 2021). Although FL optimization methods ensure that private data is never directly transmitted, Boenisch et al. (2023) demonstrated that the global models produced through FL can still enable the reconstruction of participants' data. Therefore, it is essential to study *differentially private* distributed optimization methods for differentially private training (Dwork et al., 2014; McMahan et al., 2018; Sun et al., 2019).

To mitigate emerging privacy risks in FL, differential privacy (DP) (Dwork et al., 2014) has become the standard for providing theoretical privacy guarantees in machine learning. DP is often enforced by a clipping operator. It bounds gradient sensitivity, allowing the addition of DP noise to the updates before communication. While gradient clipping enables DP as in Differentially Private Stochastic Gradient Descent (DP-SGD) (Abadi et al., 2016), it also introduces a bias that can impede convergence (Chen et al., 2020; Koloskova et al., 2023). Often, distributed DP gradient methods with clipping have been studied under assumptions that are unrealistic for heterogeneous FL environments, such as bounded gradient norms (Li et al., 2022; Wang et al., 2023; Lowy et al., 2023; Zhang et al., 2020b) or effectively ignoring the impact of clipping bias. To our knowledge, convergence guarantees for distributed DP methods remain elusive unless the impact of clipping bias is explicitly considered.

Error Feedback (EF), also known as Error Compensation (EC), such as EF21 (Richtárik et al., 2021) has been employed to alleviate the clipping bias and achieve strong convergence for non-private distributed methods with gradient clipping, as shown by Khirirat et al. (2023); Yu et al. (2023). However, extending these methods to the private setting remains an open problem. Furthermore, optimizing the convergence of distributed DP clipping methods is challenging because the clipping threshold significantly influences both the convergence speed and DP noise variance. Extensive grid search for the optimal clipping threshold is computationally expensive (Andrew et al., 2021) and leads to additional privacy loss (Papernot & Steinke, 2022). Two major approaches have emerged

| Property | DP-SGD | Clip21 | $\alpha$-NormEC (**Ours**) |
|---|---|---|---|
| Non-private Convergence | ✗ | ✓ | ✓ |
| Private Convergence | ✗ | ✗ | ✓ |
| Operators | Clipping / Normalization | Clipping | Normalization |
| Additional Assumptions | Bounded gradient or heterogeneity | No | No |

Table 1: Comparison of *distributed* private optimization methods. Unlike prior work, which either lacks private convergence guarantees or requires restrictive assumptions like bounded gradients, $\alpha$-NormEC is the first to provide these guarantees under standard smoothness conditions alone.

to address the need to manually tune the clipping threshold. The first is to use adaptive clipping techniques, such as adaptive quantile clipping, initially proposed by Andrew et al. (2021) and further analyzed by Merad & Gaïffas (2024); Shulgin & Richtárik (2024). The second, which is the focus of this paper, is to replace clipping with a normalization operator.

Smoothed normalization, introduced by Bu et al. (2024); Yang et al. (2022), is the normalization operator that offers an alternative to clipping. Unlike clipping, smoothed normalization eliminates the need to tune the clipping threshold. By ensuring that the Euclidean norm of the normalized gradient is bounded above by one, smoothed normalization guarantees robust performance of DP-SGD in private setting. However, very limited literature characterizes properties of smoothed normalization and a rigorous convergence analysis for DP-SGD using this operator, especially in the distributed setting. While the method has been studied in the single-node setting, the convergence results rely on unrealistic and/or restrictive assumptions, such as symmetric gradient noise (Bu et al., 2024) and almost sure bounds on the gradient noise variance (Yang et al., 2022).

### 1.1 CONTRIBUTIONS

We propose $\alpha$-NormEC, a distributed gradient method leveraging smoothed normalization and error compensation. It is the first algorithm to achieve provable convergence for DP, non-convex distributed optimization without relying on restrictive assumptions. Our key contributions are:

• **Favorable properties of smoothed normalization.** In Section 2.1, we show that smoothed normalization exhibits a contractive-like property similar to biased compression operators (Beznosikov et al., 2023; Shulgin & Richtárik, 2022). This property combined with **novel**, induction-based proof technique essentially allows analyzing $\alpha$-NormEC without ignoring the impact of bias.

• **Convergence for non-convex, smooth problems without restrictive assumptions.** In Section 3, we prove that $\alpha$-NormEC achieves the optimal convergence rate (Carmon et al., 2020) for minimizing non-convex, smooth functions without imposing additional restrictive assumptions, such as bounded gradient norms or bounded heterogeneity. Furthermore, $\alpha$-NormEC achieves a faster convergence rate than Clip21 (Khirirat et al., 2023) and does not require inaccessible knowledge of value $f(x^0) - f^{\inf}$.

• **The first provable convergence in the private setting.** In Section 4, we extend $\alpha$-NormEC to the DP setting. Specifically, $\alpha$-NormEC achieves the first convergence guarantees for DP, non-convex, smooth problems *without* ignoring the bias introduced by smoothed normalization. This is the first provably efficient distributed method in the DP setting under standard assumptions, thus addressing the gap left by prior work (Khirirat et al., 2023). Theoretical comparisons between DP-SGD, Clip21, and $\alpha$-NormEC are summarized in Table 1.

• **Robust empirical performance of $\alpha$-NormEC.** In Section 5, we verify the theoretical benefits of $\alpha$-NormEC in both non-private and private training via experiments. Our algorithm demonstrates robust convergence across different parameter values and benefits from error compensation that enables superior performance over vanilla distributed gradient normalization methods (such as DP-SGD). In the private training, server normalization enhances the robustness of DP-$\alpha$-NormEC across tuning parameters. Finally, DP-$\alpha$-NormEC without server normalization outperforms DP-Clip21.

### 1.2 RELATED WORK

**Clipping and normalization.** Clipping and normalization address many key challenges in machine learning. They mitigate the problem of exploding gradients in recurrent neural networks (Pascanu

et al., 2013), enhance neural network training for tasks in natural language processing (Merity et al., 2018; Brown et al., 2020) and computer vision (Brock et al., 2021), enable differentially private machine learning (Abadi et al., 2016; McMahan et al., 2018), and provide robustness in the presence of misbehaving or adversarial workers (Karimireddy et al., 2021; Özfatura et al., 2023; Malinovsky et al., 2023). In this paper, we consider smoothed normalization, introduced by Yang et al. (2022); Bu et al. (2024), as an alternative to clipping, given its robust empirical performance and hyperparameter tuning benefits in the DP setting.

**Private optimization methods.** DP-SGD (Abadi et al., 2016) is the standard distributed first-order method that achieves the DP guarantee by clipping the gradient before adding noise scaled with the clipped gradient's sensitivity. However, existing DP-SGD convergence analyses often neglect the clipping bias. Specifically, convergence results for smooth functions under differential privacy often require either the assumption of bounded gradient norms (Zhang et al., 2020b; Li et al., 2022; Zhang et al., 2022; Wang et al., 2023; Lowy et al., 2023; Murata & Suzuki, 2023; Wang et al., 2024) or conditions where clipping is effectively inactive (Zhang et al., 2024; Noble et al., 2022). Thus, the convergence behavior of DP-SGD in the presence of clipping bias remains poorly understood.

**Single-node non-private methods with clipping.** The impact of clipping bias has been extensively studied in single-node gradient methods for non-private optimization. Numerous works have shown strong convergence guarantees of clipped gradient methods under various conditions, including nonsmooth, rapidly growing convex functions (Shor, 2012; Ermoliev, 1988; Alber et al., 1998), generalized smoothness (Zhang et al., 2020a; Koloskova et al., 2023; Gorbunov et al., 2025; Vankov et al., 2025; Lobanov et al., 2024; Hübler et al., 2024), and heavy-tailed noise (Gorbunov et al., 2020a; Nguyen et al., 2023; Gorbunov et al., 2024; Hübler et al., 2025; Chezhegov et al., 2024).

**Distributed non-private methods with clipping.** Applying gradient clipping in the distributed setting is challenging. Existing convergence analyses often rely on bounded heterogeneity assumptions, which often do not hold in cases of arbitrary diverse clients. For example, federated optimization methods with clipping have been analyzed under the bounded difference between the local and global gradients (Wei et al., 2020; Liu et al., 2022; Crawshaw et al., 2023; Li et al., 2024). However, even in the non-private setting, these distributed clipping methods do not converge for simple problems (Chen et al., 2020; Khirirat et al., 2023) for arbitrary clipping threshold. To address the convergence issue, one approach is to use error feedback mechanisms, such as EF21 (Richtárik et al., 2021), that are employed by Khirirat et al. (2023); Yu et al. (2023) to compute local gradient estimators and alleviate clipping bias. However, these distributed clipping methods using error feedback are limited to the non-private setting, and extending them to the DP setting is still an open problem. In this paper, we propose a distributed method that replaces clipping with smoothed normalization in the EF21 mechanism. Our method provides the first provable convergence in the DP setting and empirically outperforms the distributed version of DP-SGD with smoothed normalization Bu et al. (2024); Yang et al. (2022), a special case of Das et al. (2022).

**Error feedback.** Error feedback, or error compensation, has been applied to improve the convergence of distributed methods with gradient compression for communication-efficient learning. First introduced by Seide et al. (2014), EF14 was extensively analyzed for first-order methods in both single-node (Stich et al., 2018; Karimireddy et al., 2019; Stich & Karimireddy, 2020; Khirirat et al., 2019) and distributed settings (Wu et al., 2018; Alistarh et al., 2018; Gorbunov et al., 2020b; Qian et al., 2021b; Tang et al., 2019; Danilova & Gorbunov, 2022; Qian et al., 2023). Another error feedback variant is EF21 proposed by Richtárik et al. (2021) that ensures strong convergence under any contractive compression operator for non-convex, smooth problems. Recent variants, e.g. EF21-SGD2M (Fatkhullin et al., 2024) and EControl (Gao et al., 2024), have been developed to obtain the lower iteration and communication complexities than EF21 for stochastic optimization.

## 2 PRELIMINARIES

We focus on solving the finite-sum optimization problem:

$$\min_{x \in \mathbb{R}^d} \left\{ f(x) := \frac{1}{n} \sum_{i=1}^{n} f_i(x) \right\}, \tag{1}$$

where $x \in \mathbb{R}^d$ is the vector of model parameters of dimension $d$, and $f_i : \mathbb{R}^d \to \mathbb{R}$ is either a loss function on client $i \in [1, n]$ (distributed setting) or data point $i$ (single-node setting). Moreover,

we impose the following assumption on objective functions commonly used for analyzing the convergence of first-order optimization algorithms (Nesterov et al., 2018).

**Assumption 1.** *Consider Problem* (1).

1. *Let $f : \mathbb{R}^d \to \mathbb{R}$ be bounded from below by a finite constant $f^{\mathrm{inf}}$, i.e. $f(x) \geq f^{\mathrm{inf}} > -\infty$ for all $x \in \mathbb{R}^d$, and be $L$-smooth, i.e. $\|\nabla f(x) - \nabla f(y)\| \leq L \|x - y\|$ for all $x, y \in \mathbb{R}^d$.*

2. *Let $f_i : \mathbb{R}^d \to \mathbb{R}$ be $L_i$-smooth, i.e. $\|\nabla f_i(x) - \nabla f_i(y)\| \leq L_i \|x - y\|$ for all $x, y \in \mathbb{R}^d$.*

## 2.1 DP-SGD

To solve Problem (1), the most common approach that ensures the approximate $(\epsilon, \delta)$-differential privacy (Dwork et al., 2006) is via the DP-SGD method (Abadi et al., 2016)

$$x^{k+1} = x^k - \gamma \Big( \frac{1}{B} \sum_{i \in \mathcal{B}^k} \Psi(\nabla f_i(x^k)) + z^k \Big), \tag{2}$$

where $\gamma > 0$ is the step size, $\mathcal{B}^k$ is a subset of $\{1, 2, \ldots, n\}$ with cardinality $|\mathcal{B}^k| = B$, $z^k \in \mathbb{R}^d$ is the DP noise, and $\Psi : \mathbb{R}^d \to \mathbb{R}^d$ is an operator with bounded norm, i.e. $\|\Psi(g)\| \leq \Phi$ for some $\Phi > 0$ and any $g \in \mathbb{R}^d$. The method (2) achieves $(\epsilon, \delta)$-DP (Abadi et al., 2016) if $z^k$ is zero-mean Gaussian noise with variance

$$\sigma_{\mathrm{DP}}^2 \geq \Phi^2 \cdot \frac{cB^2}{n^2} \frac{K \log(1/\delta)}{\epsilon^2}, \tag{3}$$

where $c > 0$ is a constant, and $K > 0$ is the total number of iterations. To obtain reasonable DP guarantees, we usually choose $\epsilon \leq 10$ and $\delta \ll 1/n$, where $n$ is the number of data points (Ponomareva et al., 2023). Notice that the DP Gaussian noise variance (3) is scaled with the *sensitivity* $\Phi$.

The method (2) has been analyzed, e.g. by Zhang et al. (2020b; 2022), under the bounded gradient norm assumption

$$\|\nabla f_i(x)\| \leq \Phi \quad \text{for all } i \text{ and } x \in \mathbb{R}^d. \tag{4}$$

However, this assumption has several limitations. First, the sensitivity $\Phi$ is typically infeasible to compute for many loss functions used in training machine learning models. Even when it can be estimated, the resulting upper bound is often overly pessimistic, leading to excessively large DP noise and thus significantly degrading the convergence performance. Second, this assumption restricts the class of models and loss functions $f$, as it excludes simple quadratic functions over unbounded domains. Third, this assumption is "pathological" in the distributed setting because it restricts the heterogeneity between different clients and can result in vacuous bounds (Khaled et al., 2020).

To enforce bounded sensitivity without imposing the bounded gradient norm, Abadi et al. (2016) suggested clipping with threshold $\tau > 0$

$$\mathrm{Clip}_\tau(g) := \min(1, \tau/\|g\|) g. \tag{5}$$

Here, the sensitivity $\Phi$ is the clipping threshold $\tau$, as $\|\mathrm{Clip}_\tau(g)\| \leq \tau = \Phi$. In general, the method (2) that uses clipping (5) is referred to as DP-SGD in the literature. However, it is challenging to analyze the convergence of DP-SGD without additional restrictive assumptions such as the symmetric noise assumption (Chen et al., 2020; Qian et al., 2021a). Even without DP noise, DP-SGD fails to converge due to the clipping bias (Koloskova et al., 2023). Furthermore, since smaller values of $\tau$ imply stronger privacy but larger bias, jointly optimizing convergence and privacy of DP-SGD by carefully tuning $\tau$ and $\gamma$ in the DP setting is a difficult task (Kurakin et al., 2022; Bu et al., 2024).

**Smoothed normalization as an alternative to clipping.** To eliminate the need to tune the clipping threshold $\tau$, smoothed normalization is an operator alternative (Bu et al., 2024; Yang et al., 2022):

$$\mathrm{Norm}_\alpha(g) := \frac{1}{\alpha + \|g\|} g, \tag{6}$$

for some $\alpha \geq 0$ and satisfies the following property.

**Lemma 1.** *For any $\alpha \geq 0$, $\beta > 0$, and $g \in \mathbb{R}^d$,*

$$\|\text{Norm}_\alpha (g)\| \leq 1, \quad and \quad \|g - \beta\text{Norm}_\alpha (g)\|^2 = \left(1 - \frac{\beta}{\alpha + \|g\|}\right)^2 \|g\|^2. \tag{7}$$

Clearly, smoothed normalization ensures that (A) the norm of the normalized vector is bounded above by 1, and (B) the distance between the true vector and a $\beta$-multiple of the normalized vector is bounded by a function of $\beta$, $\alpha$, and $\|g\|$. Although smoothed normalization with $\alpha = 0$ recovers standard normalization $g/\|g\|$ (Nesterov, 1984; Hazan et al., 2015; Levy, 2016), smoothed normalization with $\alpha > 0$ improves the contraction factor, compared to standard normalization. Specifically, as $\|g\| \to 0$, the contraction factor of smoothed normalization approaches $(1 - \beta/\alpha)^2$, while standard normalization lacks this contraction property.

Although DP-SGD (2) with smoothed normalization achieves robust empirical empirical performance in the DP setting (Bu et al., 2024), its theoretical convergence is limited to the single-node setting and relies on restrictive assumptions like the central symmetry of stochastic gradients.

## 2.2 Limitations of DP Distributed Gradient Methods

Extending the convergence results of DP-SGD to the distributed setting poses significant challenges due to potential client heterogeneity. Existing results often address the bias introduced by the operator (clipping or normalization) by relying on restrictive assumptions, such as imposing bounded gradient norms (Li et al., 2022; Zhang et al., 2022; Murata & Suzuki, 2023; Wang et al., 2024), or assuming that clipping is effectively turned off (Zhang et al., 2024; Noble et al., 2022). Recently, Li et al. (2024) extended the analysis of Koloskova et al. (2023) to a distributed private setting under strong gradient dissimilarity condition. However, their method fails to converge due to the clipping bias, as discussed in the previous section. More importantly, even in the absence of the DP noise ($z^k = 0$), the inherent bias in the gradient estimator can severely impact the convergence. For instance, DP-SGD (2) diverge exponentially when $\Psi(\cdot)$ is a Top-1 compressor (Beznosikov et al., 2023), and fail to converge when $\Psi(\cdot)$ is clipping (Chen et al., 2020; Khirirat et al., 2023).

Also, DP-SGD that uses smoothed normalization (6) directly fails to converge, as shown in the example below:

**Example 1.** *Consider Problem (1) with $n = 2, d = 1$, $f_1(x) = \frac{1}{2}(x - 3)^2$ and $f_2(x) = \frac{1}{2}(x + 3)^2$. Then $f(x) = \frac{1}{2}(f_1(x) + f_2(x))$ satisfies Assumption 1 and is minimized at $x^\star = 0$. The iterates $\{x^k\}$ generated by (2) (for $B = 2$) with $z^k = 0$ and $\alpha = 0$ do not progress when $x^0 = 2$, as the gradient estimator $\text{Norm}_\alpha \left(\nabla f_1(x^k)\right) + \text{Norm}_\alpha \left(\nabla f_2(x^k)\right)$ results in*

$$\frac{\nabla f_1(x^0)}{\|\nabla f_1(x^0)\|} + \frac{\nabla f_2(x^0)}{\|\nabla f_2(x^0)\|} = -1/1 + 5/5 = 0.$$

Naively applying normalization to the gradients in DP-SGD leads to the method that does not converge in the distributed setting without further assumptions. This fundamental limitation affects federated averaging algorithms that apply normalization on the client updates (Das et al., 2022).

## 2.3 EF21 Mechanism

One approach to resolve the convergence issues of distributed gradient methods with biased operators is to use EF21, an error feedback mechanism developed by Richtárik et al. (2021). Instead of directly applying the biased gradient estimator $\Psi$ to the gradient, EF21 applies $\Psi$ to the *difference* between the true gradient and the current error feedback (memory) vector. At iteration $k = 0, \dots, K$, each client $i$ receives the current iterate $x^k$ from the central server, and computes its local update $g_i^{k+1}$ via

$$g_i^{k+1} = g_i^k + \beta\Psi(\nabla f_i(x^k) - g_i^k), \tag{8}$$

where $\beta > 0$. Next, the central server receives the average of local error feedback vectors that are communicated by all clients $(1/n) \sum_{i=1}^n \Psi(\nabla f_i(x^k) - g_i^k)$, computes the global gradient estimator

$$g^{k+1} = g^k + \frac{\beta}{n} \sum_{i=1}^n \Psi(\nabla f_i(x^k) - g_i^k), \tag{9}$$

and updates the next iterate via $x^{k+1} = x^k - \gamma g^{k+1}$. This method generalizes EF21 by replacing a contractive compressor[1] with other biased estimators, such as clipping in Clip21 proposed by Khirirat et al. (2023). Although Clip21 attains the $\mathcal{O}(1/\sqrt{K})$ convergence in the non-private setting, deriving its convergence in the presence of the DP noise is a challenging task. As clipping does not satisfy the contractive property similar to contractive compressors required by EF21 (Appendix A.3), its convergence analysis relies heavily on separate descent inequalities when clipping is turned on and off (Appendix A.4). Also, the clipping threshold $\tau$ intricately influences both privacy and convergence guarantees. A sufficiently large $\tau$ is needed to achieve the descent inequality, but this condition requires adding large Gaussian noise, which may in turn prevent the convergence when it is accumulated.

## 3  $\alpha$-NORMEC IN THE NON-PRIVATE SETTING

To address the convergence challenges of Clip21, we propose $\alpha$-NormEC presented in Algorithm 1 Note that $\alpha$-NormEC recovers EF21 and Clip21, when we replace smoothed normalization with contractive compressor and clipping, respectively. Additionally we employ *Server Normalization* (SN) $x^{k+1} = x^k - \gamma \hat{g}^{k+1}/\|\hat{g}^{k+1}\|$ and adopt notation $0/0 = 0$. In the main part we focus on Algorithm 1, and in Appendix B.5 variation of $\alpha$-NormEC without SN is analyzed. Our experiments (in Appendix D.3) indicate that, while $\alpha$-NormEC with SN offers robust performance, the variant without it achieves fastest convergence.

---

**Algorithm 1** (DP-)$\alpha$-NormEC

---

1: **Input:** Step size $\gamma > 0$; $\beta > 0$; normalization parameter $\alpha > 0$; initial points $x^0, g_i^0 \in \mathbb{R}^d$ for $i \in [1, n]$; $\hat{g}^0 = \frac{1}{n} \sum_{i=1}^n g_i^0$; $z_i^k \in \mathbb{R}^d$ are sampled from Gaussian distribution with zero mean and $\sigma_{\text{DP}}^2$-variance.
2: **for** each iteration $k = 0, 1, \ldots, K$ **do**
3:     **for** each client $i = 1, 2, \ldots, n$ in parallel **do**
4:         Compute local gradient $\nabla f_i(x^k)$
5:         Compute $\Delta_i^k = \text{Norm}_\alpha \left( \nabla f_i(x^k) - g_i^k \right)$
6:         Update $g_i^{k+1} = g_i^k + \beta \Delta_i^k$
7:         **Non-private setting:** Transmit $\hat{\Delta}_i^k = \Delta_i^k$
8:         **Private setting:** Transmit $\hat{\Delta}_i^k = \Delta_i^k + z_i^k$
9:     **end for**
10:    Server computes $\hat{g}^{k+1} = \hat{g}^k + (\beta/n) \sum_{i=1}^n \hat{\Delta}_i^k$
11:    Server updates $x^{k+1} = x^k - \gamma \hat{g}^{k+1}/\|\hat{g}^{k+1}\|$
12: **end for**
13: **Output:** $x^{K+1}$

---

In the non-private setting, $\alpha$-NormEC has a simpler convergence analysis and stronger guarantees than Clip21. In the DP setting, $\alpha$-NormEC is the first to achieve provable convergence under standard assumptions. These advantages result from the contractive-like property of smoothed normalization, which distinguishes it from clipping (see Appendix A.3). In contrast to EF21, which relies on the strong contractivity of its compressors, $\alpha$-NormEC leverages the monotonicity of smoothed normalization to obtain the descent inequality.

The first theorem presents the convergence of $\alpha$-NormEC in the non-private setting.

**Theorem 1** (Non-private setting). *Consider $\alpha$-NormEC (Algorithm 1) for solving Problem (1), where Assumption 1 holds. Let $\beta, \alpha, \gamma > 0$ be chosen such that*

$$\frac{\beta}{\alpha + R} < 1, \quad and \quad \gamma \leq \frac{\beta R}{\alpha + R} \frac{1}{L_{\max}},$$

*where $R = \max_{i \in [1,n]} \left\| \nabla f_i(x^0) - g_i^0 \right\|$ and $L_{\max} = \max_{i \in [1,n]} L_i$. Then,*

$$\min_{k \in [0,K]} \left\| \nabla f(x^k) \right\| \leq \frac{f(x^0) - f^{\inf}}{\gamma(K+1)} + 2R + \frac{L}{2}\gamma.$$

---

[1]A contractive compressor (Stich et al., 2018) is defined by $\|g - \mathcal{C}(g)\|^2 \leq (1 - \eta)^2 \|g\|^2$ for $\eta \in (0, 1]$.

From Theorem 1, $\alpha$-NormEC converges sublinearly up to the additive constant of $2R + \frac{L}{2}\gamma$. By proper choices of parameters, $\alpha$-NormEC attains (details in Appendix B.2):

$$\min_{k\in[0,K]} \left\|\nabla f(x^k)\right\| \leq \frac{\sqrt{2L(f(x^0) - f^{\text{inf}})} + 2D}{(K+1)^{1/2}}. \tag{10}$$

The first term $\sqrt{2L(f(x^0) - f^{\text{inf}})}(K+1)^{-1/2}$ matches classical gradient descent (Carmon et al., 2020), while the second term $2D(K+1)^{-1/2}$ comes from initializing $x^0, g_i^0 \in \mathbb{R}^d$ such that $R = \max_{i\in[1,n]} \left\|\nabla f_i(x^0) - g_i^0\right\| = D(K+1)^{-1/2}$ with $D > 0$. The upper-bound for $R$ can be ensured by, for instance, setting $g_i^0 = \nabla f_i(x^0) + e$, where $e = (D/\sqrt{K+1}, 0, \dots, 0) \in \mathbb{R}^d$.

**Comparison to Clip21.** In the non-private setting, $\alpha$-NormEC has a simpler proof and provides better convergence guarantees than Clip21. Specifically, the convergence bound of $\alpha$-NormEC in (10) exhibits a smaller factor than that of Clip21, as explained in Appendix B.3. Furthermore, the hyperparameters of $\alpha$-NormEC ($\beta, \alpha, \gamma$), according to Theorem 1, are easier to implement. In particular, the step-size $\gamma$ for $\alpha$-NormEC avoids reliance on the practically inaccessible sub-optimality gap $f(x^0) - f(x^\star)$, in contrast to Clip21, whose step-size (Theorem 5.6 of Khirirat et al. (2019)) also depends on $\max_{i\in[1,n]} \|\nabla f_i(x^0)\|$.

**Comparison to EF21.** Although $\alpha$-NormEC incorporates Error Compensation similarly to EF21, these two methods rely on fundamentally different operator conditions and proof techniques. EF21 requires a strong *fixed* contractivity condition of compressors. Smoothed normalization violates this assumption as it satisfies *state-dependent* contractive-like property (7). That is why we rely on a novel induction-based proof for $\alpha$-NormEC as detailed in Appendix B.4.

**Extensions of $\alpha$-NormEC.** Our analysis can be extended to establish the convergence of $\alpha$-NormEC without server normalization in Appendix B.5, and $\alpha$-NormEC using stochastic local gradients in Appendix B.6.

## 4 $\alpha$-NORMEC IN THE DP SETTING

Next, we demonstrate that $\alpha$-NormEC achieves provable convergence guarantees in the DP setting, the feature that Clip21 lacks. As shown in Algorithm 1, each client applies smoothed normalization to its gradient before injecting DP noise. Because smoothed normalization ensures the sensitivity is always 1, we can prove that $\alpha$-NormEC achieves DP and utility guarantees by appropriately choosing the variance of the DP noise according to Abadi et al. (2016).

To show this, we present the convergence of DP-$\alpha$-NormEC next.

**Theorem 2** (DP setting). *Consider* DP-$\alpha$-NormEC *(Algorithm 1) for solving Problem (1), where Assumption 1 holds. Let $\beta, \alpha, \gamma > 0$ be chosen such that*

$$\frac{\beta}{\alpha + R} < 1, \quad \text{and} \quad \gamma \leq \frac{\beta R}{\alpha + R}\frac{1}{L_{\max}},$$

*where $R = \max_{i\in[1,n]} \left\|\nabla f_i(x^0) - g_i^0\right\|$, and $L_{\max} = \max_{i\in[1,n]} L_i$. Then,*

$$\min_{k\in[0,K]} \mathrm{E}\left[\left\|\nabla f(x^k)\right\|\right] \leq \frac{f(x^0) - f^{\text{inf}}}{\gamma(K+1)} + 2R + \frac{L}{2}\gamma + 2\sqrt{\frac{\beta^2(K+1)\sigma_{\text{DP}}^2}{n}}.$$

In the DP setting, from Theorem 2, $\alpha$-NormEC achieves the sublinear convergence up to the additive term $2R + \gamma L/2 + 2\sqrt{\beta^2(K+1)\sigma_{\text{DP}}^2/n}$. Notice that $\alpha$-NormEC in the DP setting introduces one additional term that arises from the DP noise $\sigma_{\text{DP}}^2$. This additive constant diminishes when we initialize memory vectors $g_i^0 \in \mathbb{R}^d$ such that $R$ becomes small and when we properly choose parameters $\gamma, \beta > 0$ (details in Appendix C.1.1). Furthermore, we can use secure aggregation techniques to initialize the memory vector $\hat{g}^0 = \frac{1}{n}\sum_{i=1}^n g_i^0$ at the server, without revealing clients' local gradients $g_i^0$. This is shown in Appendix C.3 by having all clients add cryptographic noise before their updates are communicated to the server.

**Utility guarantees.** Unlike Clip21 (Khirirat et al., 2023), $\alpha$-NormEC achieves $(\epsilon, \delta)$-DP[2] and comes with convergence guarantees. We show this by setting the standard deviation of the DP noise

---

[2]Privacy guaranty follows from Theorem 1 by Abadi et al. (2016).

according to Theorem 1 of Abadi et al. (2016), i.e., $\sigma_{\text{DP}} = \mathcal{O}(\sqrt{(K+1)\log(1/\delta)}\epsilon^{-1})$, which yields the utility bound $\mathcal{O}\left(\Delta\sqrt[4]{\frac{d\log(1/\delta)}{n\epsilon^2}}\right)$ with constant $\Delta = \sqrt{L_{\max}(f(x^0) - f^{\inf})}$ (further details in Corollary 2). Unlike Clip21, $\alpha$-NormEC provides the first utility bound in the DP distributed setting that accounts for the effect of bounding sensitivity, a factor often neglected in the existing literature. Our obtained utility bound applies for smooth problems without the bounded gradient norm assumption, the limitation present in prior works for analyzing DP-SGD, such by Li et al. (2022); Wang et al. (2023); Lowy et al. (2023); Zhang et al. (2020b).

## 5 EXPERIMENTS

We evaluate the performance of $\alpha$-NormEC for deep neural network training in both non-private and private settings. We conduct experiments on the CIFAR-10 (Krizhevsky et al., 2009) dataset using the ResNet20 (He et al., 2016) model for the image classification task. The compared methods are run for 300 communication rounds. The convergence plots present results for tuned step size $\gamma$. Additional experimental details and results are provided in Appendix D.

### 5.1 NON-PRIVATE TRAINING

$\alpha$-NormEC **demonstrates stable convergence across the normalization parameter ($\alpha$) values $\alpha$, and robustness across $\beta$ values.** From Figure 1, we observe that convergence of $\alpha$-NormEC is stable with respect to a wide range of $\alpha$ values and robust to variations in $\beta$. The performance of $\alpha$-NormEC is primarily governed by the choice of $\beta$. From Figure 1, optimal performance (85-86% accuracy) is observed when $\beta$ is around 0.1. While $\alpha$-NormEC is stable with respect to $\alpha$, extreme values of $\beta$ lead to suboptimal performance: very large values ($\beta = 10.0$) result in lower accuracy (81-82%), while very small values ($\beta = 0.01$) achieve moderate per-

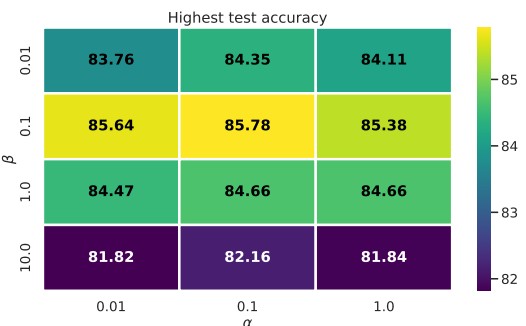

Figure 1: The highest test accuracy achieved by $\alpha$-NormEC with different $\alpha$ and $\beta$ values.

formance (83-84%). The optimal configuration, achieving the highest 85.78% accuracy, is $\beta = 0.1$ and $\alpha = 0.1$. Further analysis of the algorithm's stability with respect to $\alpha$ and robustness to $\beta$, including additional metrics (along with convergence curves) is provided in Appendix D.2.1. For subsequent experiments, we set $\alpha = 0.01$, which is consistent with recommendations from prior work in the single-node setting (Bu et al., 2024).

**Error compensation enables $\alpha$-NormEC to outperform DP-SGD.** From Figure 2, $\alpha$-NormEC outperforms DP-SGD with smoothed normalization (defined by Equation(2) with $B = n$ and $z \equiv 0$). This improvement is attributed to error compensation (EC), as confirmed by running $\alpha$-NormEC without server normalization (Line 11 of Algorithm 1). From Figure 2, $\alpha$-NormEC achieves faster convergence than DP-SGD with smoothed normalization for most $\beta$ values, with the exception of $\beta = 10$. However, such a large $\beta$ is impractical for differentially private training due to the resulting increase in noise variance. Moreover, while our algorithm demonstrates robust performance across varying $\beta$ values, DP-SGD with smoothed normalization exhibits greater sensitivity to this parameter, notably struggling with convergence at $\beta = 0.01$. This comparison underscores how EC not only accelerates convergence but also improves the algorithm's stability with respect to parameter selection. Appendix D.2.2 presents further details (such as accuracy convergence curves in Figure 6) and optimal parameters with corresponding final accuracies (Table 7).

An ablation study examining the impact of server normalization is provided in Appendix D.2.3. Furthermore, a comparison between $\alpha$-NormEC and Clip21 is presented in Appendix D.2.4.

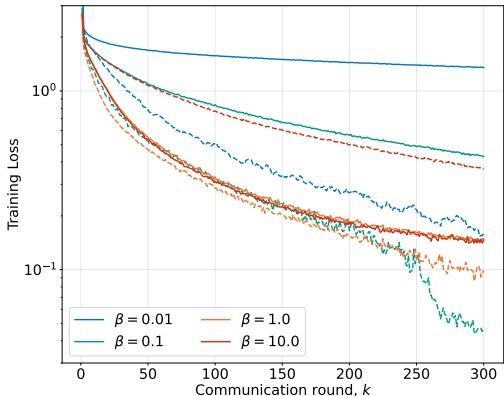
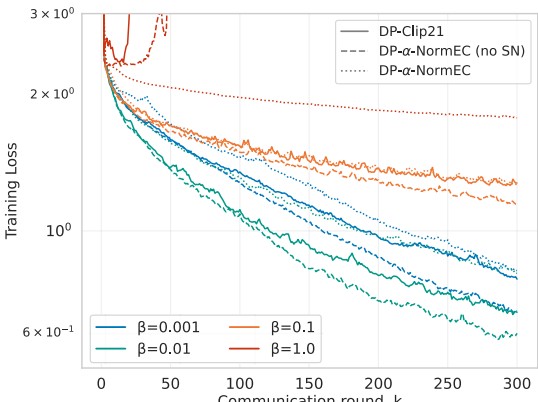

Figure 2: Comparison of DP-SGD (2) [solid] and $\alpha$-NormEC (1) [dashed] without server normalization.

Figure 3: Comparison of methods in the Differentially Private (DP) setting across different $\beta$ values.

## 5.2 PRIVATE TRAINING

We analyze the performance of $\alpha$-NormEC in the differentially private setting by setting the variance of added noise at $\beta\sqrt{K\log(1/\delta)}\epsilon^{-1}$ for $\epsilon = 8, \delta = 10^{-5}$ and vary $\beta$ to simulate different privacy levels. Figure 3 shows the training loss curves for DP-$\alpha$-NormEC (with and without server normalization) and DP-Clip21[3]. Notably, compared to the non-private case, convergence in the DP setting is slower and requires a smaller $\beta$ (e.g., 0.01) for best performance.

From Figure 3 we observe three key findings: **(1)** DP-$\alpha$-NormEC without Server Normalization converges significantly faster than DP-Clip21 at all privacy levels ($\beta \in \{0.001, 0.01, 0.1\}$); **(2)** Server normalization (SN) provides crucial stability at high noise levels at $\beta = 1.0$, only DP-$\alpha$-NormEC with SN maintains convergence; **(3)** While SN improves robustness, it comes with a slight reduction in convergence speed at lower noise levels.

The complete analysis, including test accuracy results across different hyperparameter combinations and detailed performance comparisons, is presented in Appendix D.3. Notably, server normalization significantly reduces performance variation across different learning rates ($\gamma$), with at most $6\%$ variation compared to $40\%$ without normalization, demonstrating improved hyperparameter robustness. The superiority of DP-$\alpha$-NormEC is further confirmed in a high-privacy regime with a stricter budget of $\epsilon = 1$ in Appendix D.3.1.

## 6 CONCLUSION

We have proposed and analyzed $\alpha$-NormEC, a novel distributed algorithm that integrates smoothed normalization with the EF21 mechanism for solving non-convex, smooth optimization problems in both non-private and private settings. Unlike Clip21, $\alpha$-NormEC achieves strong convergence guarantees that almost match those of classical gradient descent for non-private training and provides the first utility bound for private training without relying on restrictive assumptions such as bounded gradient norms. In neural network training, $\alpha$-NormEC achieves robust convergence across varying hyperparameters and significantly stronger convergence (due to error compensation) compared to DP-SGD with smoothed normalization. In the private training, DP-$\alpha$-NormEC benefits from server normalization for increased robustness and outperforms DP-Clip21.

Our work implies many promising research directions. One direction is to extend $\alpha$-NormEC to accommodate the partial participation case, where the central server receives the updates from a subset of clients. Another important direction is to modify $\alpha$-NormEC to solve federated learning problems, where the clients run their local updates before the local updates are normalized and transmitted to the central server.

---

[3]DP-Clip21, unlike Clip21, does not have theoretical convergence guarantees.

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

## CONTENTS

## A  PRELIMINARIES

**Notation.**  We use $[a, b]$ to denote the set $\{a, a + 1, a + 2, \dots, b\}$ for integers $a, b$ such that $a \leq b$, and $\mathrm{E}[u]$ to represent the expectation of a random variable $u$. For vectors $x, y \in \mathbb{R}^d$, $\langle x, y \rangle$ denotes their inner product, and $\|x\| := \sqrt{\langle x, x \rangle}$ denotes the Euclidean norm of $x$. Finally, for functions $f, g : \mathbb{R}^d \to \mathbb{R}$, we write $f(x) = \mathcal{O}(g(x))$ if $f(x) \leq M \cdot g(x)$ for some $M > 0$.

### A.1  BASIC FACTS

For $n \in \mathbb{N}$ and $x_1, \dots, x_n, x, y \in \mathbb{R}^d$,

$$\langle x, y \rangle \quad \leq \quad \|x\|\,\|y\|, \tag{11}$$

$$\|x + y\| \quad \leq \quad \|x\| + \|y\|, \quad \text{and} \tag{12}$$

$$\left\| \frac{1}{n} \sum_{i=1}^{n} x_i \right\| \quad \leq \quad \frac{1}{n} \sum_{i=1}^{n} \|x_i\|. \tag{13}$$

### A.2  PROOF OF LEMMA 1

We prove the first statement by taking the Euclidean norm. Next, we prove the second statement. From the definition of the Euclidean norm,

$$\|g - \beta \mathrm{Norm}_\alpha (g)\|^2 \quad \overset{(6)}{=} \quad \|g\|^2 + \frac{\beta^2}{(\alpha + \|g\|)^2} \|g\|^2 - 2\beta \frac{\|g\|^2}{\alpha + \|g\|}$$

$$= \quad \left( 1 - \frac{\beta}{\alpha + \|g\|} \right)^2 \|g\|^2.$$

### A.3  CONTRACTIVE COMPRESSION, CLIPPING AND NORMALIZATION

We summarize the properties of three key biased operators: contractive compressors, clipping, and smoothed normalization in Table 2. Unlike clipping, smoothed normalization ensures the contractive-like property similar to contractive compressors.

| Operator | Property |
|---|---|
| Contractive compressor $\mathcal{C} : \mathbb{R}^d \to \mathbb{R}^d$ | $\|\mathcal{C}(g) - g\|^2 \leq (1 - \eta)^2 \|g\|^2$ |
| Clipping $\text{Clip}_\tau(g) := \min\left(1, \frac{\tau}{\|g\|}\right) g$ | $\|\text{Clip}_\tau(g) - g\|^2 \leq \max(0, \|g\| - \tau)^2$ |
| Smoothed normalization $\text{Norm}_\alpha(g) := \frac{1}{\alpha + \|g\|} g$ | $\|\text{Norm}_\alpha(g) - g\|^2 \leq \left(1 - \frac{1}{\alpha + \|g\|}\right)^2 \|g\|^2$ |

Table 2: Comparisons of contractive compressors, clipping, and smoothed normalization with their properties. Smoothed normalization, unlike clipping, satisfies the contractive property similar to compressors.

### A.4 COMPARISON OF EF21 WITH CLIPPING AND SMOOTHED NORMALIZATION

We compare the modified EF21 mechanism, where a contractive compressor is replaced with clipping in Clip21 and with smoothed normalization in $\alpha$-NormEC. To compare these modified updates, given the optimal vector $g^\star \in \mathbb{R}^d$, we consider the single-node EF21 mechanism, which computes the memory vector $g^k \in \mathbb{R}^d$ according to

$$g^{k+1} = g^k + \Psi(g^\star - g^k), \tag{14}$$

where $\Psi : \mathbb{R}^d \to \mathbb{R}^d$ is the biased gradient estimator, and $g^0 \in \mathbb{R}^d$ is the initial memory vector.

If $\Psi(g) = \text{Clip}_\tau(g)$, then from Theorem 4.3 of Khirirat et al. (2023)

$$\left\|g^k - g^\star\right\| \leq \max(0, \left\|g^0 - g^\star\right\| - k\tau).$$

If $\Psi(g) = \text{Norm}_\alpha(g)$, then from Lemma 1

$$
\begin{aligned}
\left\|g^\star - g^k\right\|^2 &= \left\|g^\star - g^{k-1} - \beta\text{Norm}_\alpha\left(g^\star - g^{k-1}\right)\right\|^2 \\
&= \left(1 - \frac{\beta}{\alpha + \|g^\star - g^{k-1}\|}\right)^2 \left\|g^\star - g^{k-1}\right\|^2 \\
&\;\;\vdots \\
&= \left\|g^\star - g^0\right\|^2 \cdot \prod_{l=1}^k \left(1 - \frac{\beta}{\alpha + \|g^\star - g^{l-1}\|}\right)^2.
\end{aligned}
$$

In conclusion, while the EF21 mechanism with clipping ensures that the memory $g^k$ will reach $g^\star$ within a finite number of iterations $k$ (when $k \geq \left\|g^0 - g^\star\right\| / \tau$), the EF21 mechanism with smoothed normalization guarantees that $g^k$ will eventually reach $g^\star$ (provided that $\beta/\alpha < 1$).

## B NON-PRIVATE RESULTS

### B.1 PROOF OF THEOREM 1

**Proof outline.** By the $L$-smoothness of the objective function $f$, and by the update for $x^{k+1}$ in $\alpha$-NormEC, we obtain

$$V^{k+1} \leq V^k - \gamma \left\|\nabla f(x^k)\right\| + \frac{L\gamma^2}{2} + 2\gamma W^k,$$

where $V^k := f(x^k) - f^{\text{inf}}$, and $W^k := \frac{1}{n}\sum_{i=1}^n \left\|\nabla f_i(x^k) - g_i^{k+1}\right\|$. The key step to establish the convergence is to bound $\left\|\nabla f_i(x^k) - g_i^{k+1}\right\|$. Using Lemma 2 and appropriate choices of the tuning parameters $\beta$, $\alpha$, and $\gamma$, we get

$$\left\|\nabla f_i(x^k) - g_i^{k+1}\right\| \leq \max_{i \in [1,n]} \left\|\nabla f_i(x^0) - g_i^0\right\|, \quad \forall k \geq 0.$$

Finally, substituting this bound into the previous descent inequality yields the convergence bound in $\min_{k \in [0,K]} \left\|\nabla f(x^k)\right\|$. Deriving the bound on $\left\|\nabla f_i(x^k) - g_i^{k+1}\right\|$ for $\alpha$-NormEC by induction is

similar to but simpler than Clip21. This simplified proof is possible because smoothed normalization possesses a contractive property similar to the contractive compressor used in EF21.

We prove Theorem 1 by Lemma 2, which states $\left\|\nabla f_i(x^{k+1}) - g_i^{k+1}\right\| \leq R$ for some positive scalars $R$, given that $\left\|\nabla f_i(x^k) - g_i^k\right\| \leq R$, and hyperparameters $\alpha, \beta, \gamma$ are properly tuned.

**Lemma 2** (Non-private setting). *Consider $\alpha$-NormEC (Algorithm 1) for solving Problem (1), where Assumption 1 holds. If $\left\|\nabla f_i(x^k) - g_i^k\right\| \leq R$, $\frac{\beta}{\alpha+R} < 1$, and $\gamma \leq \frac{\beta R}{\alpha+R}\frac{1}{L_{\max}}$ with $L_{\max} = \max_{i\in[1,n]} L_i$, then $\left\|\nabla f_i(x^{k+1}) - g_i^{k+1}\right\| \leq R$.*

*Proof.* From the definition of the Euclidean norm,

$$
\begin{aligned}
\left\|\nabla f_i(x^{k+1}) - g_i^{k+1}\right\| \quad &\overset{(12)}{\leq} \quad \left\|\nabla f_i(x^{k+1}) - \nabla f_i(x^k)\right\| + \left\|\nabla f_i(x^k) - g_i^{k+1}\right\| \\
&\overset{g_i^{k+1}}{=} \quad \left\|\nabla f_i(x^{k+1}) - \nabla f_i(x^k)\right\| \\
&\qquad + \left\|\nabla f_i(x^k) - g_i^k - \beta\mathrm{Norm}_\alpha\left(\nabla f_i(x^k) - g_i^k\right)\right\| \\
&\overset{\text{Lemma 1}}{\leq} \quad \left\|\nabla f_i(x^{k+1}) - \nabla f_i(x^k)\right\| \\
&\qquad + \left|1 - \frac{\beta}{\alpha + \left\|\nabla f_i(x^k) - g_i^k\right\|}\right| \left\|\nabla f_i(x^k) - g_i^k\right\| \\
&\overset{\text{Assumption 1, and } x^{k+1}}{\leq} \quad L_{\max}\gamma + \left|1 - \frac{\beta}{\alpha + \left\|\nabla f_i(x^k) - g_i^k\right\|}\right| \left\|\nabla f_i(x^k) - g_i^k\right\|.
\end{aligned}
$$

If $\left\|\nabla f_i(x^k) - g_i^k\right\| \leq R$ and $\frac{\beta}{\alpha+R} < 1$, then $\left\|\nabla f_i(x^{k+1}) - g_i^{k+1}\right\| \leq R$ when

$$
\gamma \leq \frac{\beta R}{\alpha + R}\frac{1}{L_{\max}}.
$$

$\square$

Now, we are ready to prove the result in Theorem 1 in four steps.

**Step 1) Prove by induction that** $\left\|\nabla f_i(x^k) - g_i^k\right\| \leq R$ **for** $R = \max_{i\in[1,n]}\left\|\nabla f_i(x^0) - g_i^0\right\|$. For $k = 0$, this is obvious. Next, let $\left\|\nabla f_i(x^l) - g_i^l\right\| \leq R$ for $R = \max_{i\in[1,n]}\left\|\nabla f_i(x^0) - g_i^0\right\|$ for $l = 0, 1, \ldots, k$. Then, if $\beta/(\alpha + R) < 1$, and $\gamma \leq \frac{\beta R}{\alpha+R}\frac{1}{L_{\max}}$, then from Lemma 2 $\left\|\nabla f_i(x^{k+1}) - g_i^{k+1}\right\| \leq R$.

**Step 2) Bound** $\left\|\nabla f_i(x^k) - g_i^{k+1}\right\|$. From the definition of the Euclidean norm,

$$
\begin{aligned}
\left\|\nabla f_i(x^k) - g_i^{k+1}\right\| \quad &\overset{g_i^{k+1}}{=} \quad \left\|\nabla f_i(x^k) - g_i^k - \beta\mathrm{Norm}_\alpha\left(\nabla f_i(x^k) - g_i^k\right)\right\| \\
&\overset{\text{Lemma 1}}{\leq} \quad \left|1 - \frac{\beta}{\alpha + \left\|\nabla f_i(x^k) - g_i^k\right\|}\right| \left\|\nabla f_i(x^k) - g_i^k\right\| \\
&\overset{(*)}{\leq} \quad \left(1 - \frac{\beta}{\alpha + R}\right) R \leq R,
\end{aligned}
$$

where we reach $(*)$ by the fact that $\left\|\nabla f_i(x^k) - g_i^k\right\| \leq R$, $\frac{\beta}{\alpha+R} < 1$, and $\gamma \leq \frac{\beta R}{\alpha+R}\frac{1}{L_{\max}}$.

**Step 3) Derive the descent inequality.** By the $L$-smoothness of $f$, by the definition of $x^{k+1}$, and by the fact that $\hat{g}^{k+1} = g^{k+1}$,

$$
\begin{aligned}
f(x^{k+1}) - f^{\inf} \quad &\leq \quad f(x^k) - f^{\inf} - \frac{\gamma}{\|g^{k+1}\|} \left\langle \nabla f(x^k), g^{k+1} \right\rangle + \frac{L\gamma^2}{2} \\
&= \quad f(x^k) - f^{\inf} - \gamma \|g^{k+1}\| + \frac{\gamma}{\|g^{k+1}\|} \left\langle \nabla f(x^k) - g^{k+1}, g^{k+1} \right\rangle + \frac{L\gamma^2}{2} \\
&\overset{(11)}{\leq} \quad f(x^k) - f^{\inf} - \gamma \|g^{k+1}\| + \gamma \|\nabla f(x^k) - g^{k+1}\| + \frac{L\gamma^2}{2} \\
&\overset{(12)}{\leq} \quad f(x^k) - f^{\inf} - \gamma \|\nabla f(x^k)\| + 2\gamma \|\nabla f(x^k) - g^{k+1}\| + \frac{L\gamma^2}{2} \\
&\overset{(13)}{\leq} \quad f(x^k) - f^{\inf} - \gamma \|\nabla f(x^k)\| + 2\gamma \frac{1}{n} \sum_{i=1}^{n} \|\nabla f_i(x^k) - g_i^{k+1}\| + \frac{L\gamma^2}{2}.
\end{aligned}
$$

Next, since $\|\nabla f_i(x^k) - g_i^{k+1}\| \leq R$ with $R = \max_{i \in [1,n]} \|\nabla f_i(x^0) - g_i^0\|$, we have

$$
f(x^{k+1}) - f^{\inf} \leq f(x^k) - f^{\inf} - \gamma \|\nabla f(x^k)\| + 2\gamma \max_{i \in [1,n]} \|\nabla f_i(x^0) - g_i^0\| + \frac{L\gamma^2}{2}.
$$

**Step 4) Finalize the convergence rate.** Finally, by re-arranging the terms of the inequality,

$$
\begin{aligned}
\min_{k \in [0,K]} \|\nabla f(x^k)\| &\leq \frac{1}{K+1} \sum_{k=0}^{K} \|\nabla f(x^k)\| \\
&\leq \frac{[f(x^0) - f^{\inf}] - [f(x^{K+1}) - f^{\inf}]}{\gamma(K+1)} + 2 \max_{i \in [1,n]} \|\nabla f_i(x^0) - g_i^0\| + \frac{L}{2}\gamma \\
&\overset{(\dagger)}{\leq} \frac{f(x^0) - f^{\inf}}{\gamma(K+1)} + 2 \max_{i \in [1,n]} \|\nabla f_i(x^0) - g_i^0\| + \frac{L}{2}\gamma,
\end{aligned}
$$

where we reach $(\dagger)$ by the fact that $f^{\inf} \geq f(x^{K+1})$.

### B.2 THE $\mathcal{O}(1/\sqrt{K})$ CONVERGENCE OF $\alpha$-NormEC IN THE NON-PRIVATE SETTING

From Theorem 2, we show that $\alpha$-NormEC achieves the $\mathcal{O}(1/\sqrt{K})$ convergence in the gradient norm, which almost matches the convergence bound by classical gradient descent.

The next corollary shows the $\mathcal{O}(1/\sqrt{K})$ rate of $\alpha$-NormEC under specific choices of initialized memory vectors $g_i^0$ and the step size $\gamma$.

**Corollary 1** (Non-private setting). *Consider $\alpha$-NormEC (Algorithm 1) for solving Problem (1) under the same setting as Theorem 1. If we choose $g_i^0 \in \mathbb{R}^d$ such that $\max_{i \in [1,n]} \|\nabla f_i(x^0) - g_i^0\| = D(K+1)^{-1/2}$ with any $D > 0$, $\gamma \leq \frac{\beta}{L_{\max}} \frac{D}{\alpha+D} \frac{1}{(K+1)^{1/2}}$, and $\alpha > \beta$, then*

$$
\min_{k \in [0,K]} \|\nabla f(x^k)\| \leq \frac{C}{(K+1)^{1/2}},
$$

*where $C = \frac{L_{\max}(\alpha+D)}{\beta D}(f(x^0) - f^{\inf}) + 2D + \frac{L}{2} \frac{\beta D}{L_{\max}(\alpha+D)}$.*

*Proof.* If $g_i^0 \in \mathbb{R}^d$ is chosen such that $\max_{i \in [1,n]} \|\nabla f_i(x^0) - g_i^0\| = \frac{D}{(K+1)^{1/2}}$ with any $D > 0$, $\gamma \leq \frac{\beta}{L_{\max}} \frac{D}{\alpha+D} \frac{1}{(K+1)^{1/2}}$, and $\beta < \alpha$, then from Theorem 1, we obtain $\gamma \leq \frac{\beta R}{\alpha+R} \frac{1}{L_{\max}}$ with $R = \max_{i \in [1,n]} \|\nabla f_i(x^0) - g_i^0\|$, and

$$
\min_{k \in [0,K]} \|\nabla f(x^k)\| \leq \frac{L_{\max}(\alpha+D)}{\beta D} \frac{f(x^0) - f^{\inf}}{(K+1)^{1/2}} + 2\frac{D}{(K+1)^{1/2}} + \frac{L}{2} \frac{\beta D}{L_{\max}(\alpha+D)} \frac{1}{(K+1)^{1/2}}.
$$

$\square$

From Corollary 1, $\alpha$-NormEC enjoys the $\mathcal{O}(1/\sqrt{K})$ convergence in the gradient norm when we choose $g_i^0$ such that $R = \mathcal{O}(1/\sqrt{K})$, and $\gamma = \mathcal{O}(\beta/\sqrt{K})$.

By further choosing $\alpha > 1$, and $\beta = \frac{L_{\max}(\alpha+D)}{D}\sqrt{\frac{2(f(x^0)-f^{\inf})}{L}}$, which ensures $\frac{L_{\max}(\alpha+D)}{\beta D}(f(x^0) - f^{\inf}) = \frac{L}{2}\frac{\beta D}{L_{\max}(\alpha+D)}$, the associated convergence bound from Corollary 1 becomes

$$\min_{k\in[0,K]}\left\|\nabla f(x^k)\right\| \leq \frac{\sqrt{2L(f(x^0)-f^{\inf})}+2D}{(K+1)^{1/2}}. \tag{15}$$

This bound comprises two terms. The first term $\sqrt{2L(f(x^0)-f^{\inf})}(K+1)^{-1/2}$ is the convergence bound obtained by classical gradient descent Carmon et al. (2020), while the second term $2D(K+1)^{-1/2}$ comes from the initialized memory vectors $g_i^0$ for running the error-feedback mechanism.

### B.3 COMPARISON TO Clip21

We show that the convergence bound of $\alpha$-NormEC (15) has a smaller factor than that of Clip21 from Theorem 5.6. of Khrirat et al. (2023).

To show this, let $\hat{x}^K$ be selected uniformly at random from a set $\{x^0, x^1, \ldots, x^K\}$. Then, from Theorem 5.6. of Khrirat et al. (2023), Clip21 achieves the following convergence bound:

$$\begin{aligned}
\min_{k\in[0,K]}\left\|\nabla f(x^k)\right\| &\leq& \mathrm{E}\left[\left\|\nabla f(\hat{x}^K)\right\|\right] \\
&\leq& \sqrt{\mathrm{E}\left[\left\|\nabla f(\hat{x}^K)\right\|^2\right]} \\
&\leq& \frac{L_{\max}(f(x^0)-f^{\inf})}{\tau(K+1)^{1/2}} + \frac{\sqrt{(1+C_1/\tau)C_2}}{(K+1)^{1/2}},
\end{aligned}$$

where $\tau > 0$ is a clipping threshold, $C_1 = \max_{i\in[1,n]}\left\|\nabla f_i(x^0)\right\|$, and $C_2 = \max(\max(L,L_{\max})(f(x^0)-f^{\inf})),C_1^2)$.

If $\tau = \frac{L_{\max}}{\sqrt{2L}}\sqrt{f(x^0)-f^{\inf}}$, then

$$\begin{aligned}
\min_{k\in[0,K]}\left\|\nabla f(x^k)\right\| &\leq& \sqrt{\frac{2L(f(x^0)-f^{\inf})}{K+1}} + \frac{\sqrt{\left(1+\frac{C_1\sqrt{2L}}{L_{\max}\sqrt{f(x^0)-f^{\inf}}}\right)C_2}}{(K+1)^{1/2}} \\
&\leq& \sqrt{\frac{2L(f(x^0)-f^{\inf})}{K+1}} \\
&& + \frac{\sqrt{C_2}+\mathcal{O}\left(\max(\sqrt{C_1}\sqrt[4]{f(x^0)-f^{\inf}},C_1^3/\sqrt{f(x^0)-f^{\inf}})\right)}{(K+1)^{1/2}}.
\end{aligned}$$

The first term in the convergence bound of Clip21 matches that of $\alpha$-NormEC as given in (10). However, the second term in the convergence bound of $\alpha$-NormEC is $D/\sqrt{K+1}$, where $D > 0$ can be made arbitrarily small. In contrast, the corresponding term for Clip21 is $C/\sqrt{K+1}$, where $C > 0$ may become significantly larger than $D$ if $x^0 \in \mathbb{R}^d$ is far from the stationary point, leading to a large value of $C_1 = \max_{i\in[1,n]}\left\|\nabla f_i(x^0)\right\|$.

### B.4 COMPARISON TO EF21

$\alpha$-NormEC modifies EF21 to support distributed optimization algorithms for training machine learning models under differential privacy. However, $\alpha$-NormEC is not a special case of EF21, as the two algorithms rely on fundamentally different biased operator conditions and convergence analyses. Specifically, $\alpha$-NormEC replaces the contractive compression operators used in EF21 with smoothed normalization. While this modification enables practical advantages in private learning, it

introduces significant analytical challenges. The convergence analysis of EF21 crucially depends on the assumption that the compression operators are contractive, satisfying the condition:

$$\psi^{k+1} \leq (1-q)\psi^k, \quad \forall q \in (0,1],$$

where $\psi^k = \left\|\nabla f_i(x^k) - g_i^k\right\|$ is the gradient estimation error. Nonetheless, this strong contractivity condition does not hold in general for smoothed normalization. Rather, we prove that smoothed normalization satisfies only a contractive-like property, which is insufficient for directly applying the convergence analysis of EF21. To overcome this, we develop a new induction-based proof technique that establishes the weaker monotonicity condition:

$$\psi^{k+1} \leq \psi^k,$$

which is sufficient to guarantee convergence within our novel analysis framework for $\alpha$-NormEC.

### B.5 ANALYSIS WITHOUT SERVER NORMALIZATION

In this section, we prove the convergence in the non-private setting for $\alpha$-NormEC without the server normalization– specifically, in the variant of Algorithm 1 where the server update in Step 11 becomes $x^{k+1} = x^k - \gamma g^{k+1}$.

To facilitate our analysis, we impose one additional assumption on the objective function.

**Assumption 2.** *Let $f : \mathbb{R}^d \to \mathbb{R}$ satisfy $f(x) - f^{\inf} \leq \Delta$ for some $\Delta > 0$ and for all $x \in \mathbb{R}^d$.*

From Assumption 2, we can bound the gradient error norm in the next lemma.

**Lemma 3.** *Consider $\alpha$-NormEC (Algorithm 1) without the server normalization for solving Problem (1), where Assumptions 1 and 2 hold. If $\left\|\nabla f_i(x^k) - g_i^k\right\| \leq R$, $\frac{\beta}{\alpha+R} < 1$, and $\gamma \leq \frac{\beta R}{\alpha+R} \frac{1}{R+\sqrt{2L\Delta}} \frac{1}{L_{\max}}$ with $L_{\max} = \max_{i \in [1,n]} L_i$, then $\left\|\nabla f_i(x^{k+1}) - g_i^{k+1}\right\| \leq R$.*

*Proof.* From the definition of the Euclidean norm,

$$
\begin{aligned}
\left\|\nabla f_i(x^{k+1}) - g_i^{k+1}\right\| &\overset{(12)}{\leq} \left\|\nabla f_i(x^{k+1}) - \nabla f_i(x^k)\right\| + \left\|\nabla f_i(x^k) - g_i^{k+1}\right\| \\
&\overset{g_i^{k+1}}{=} \left\|\nabla f_i(x^{k+1}) - \nabla f_i(x^k)\right\| \\
&\quad + \left\|\nabla f_i(x^k) - g_i^k - \beta \mathrm{Norm}_\alpha\left(\nabla f_i(x^k) - g_i^k\right)\right\| \\
&\overset{\text{Lemma 1}}{\leq} \left\|\nabla f_i(x^{k+1}) - \nabla f_i(x^k)\right\| \\
&\quad + \left|1 - \frac{\beta}{\alpha + \left\|\nabla f_i(x^k) - g_i^k\right\|}\right| \left\|\nabla f_i(x^k) - g_i^k\right\| \\
&\overset{\text{Assumption 1, and } x^{k+1}}{\leq} L_{\max}\gamma \left\|\hat{g}^k\right\| \\
&\quad + \left|1 - \frac{\beta}{\alpha + \left\|\nabla f_i(x^k) - g_i^k\right\|}\right| \left\|\nabla f_i(x^k) - g_i^k\right\|.
\end{aligned}
$$

Next, since

$$
\begin{aligned}
\left\|\hat{g}^k\right\| &\overset{(12)}{\leq} \left\|\nabla f(x^k) - \hat{g}^k\right\| + \left\|\nabla f(x^k)\right\| \\
&\overset{(12)}{\leq} \frac{1}{n}\sum_{i=1}^n \left\|\nabla f_i(x^k) - \hat{g}_i^k\right\| + \left\|\nabla f(x^k)\right\| \\
&\overset{\text{Assumption 1}}{\leq} \frac{1}{n}\sum_{i=1}^n \left\|\nabla f_i(x^k) - \hat{g}_i^k\right\| + \sqrt{2L[f(x^k) - f^{\inf}]} \\
&\overset{\text{Assumption 2}}{\leq} \frac{1}{n}\sum_{i=1}^n \left\|\nabla f_i(x^k) - \hat{g}_i^k\right\| + \sqrt{2L\Delta},
\end{aligned}
$$

by the fact that $\hat{g}^k = g^k$, we obtain

$$
\left\| \nabla f_i(x^{k+1}) - g_i^{k+1} \right\| \;\leq\; \frac{L_{\max}\gamma}{n} \sum_{i=1}^{n} \left\| \nabla f_i(x^k) - g_i^k \right\| + L_{\max}\gamma\sqrt{2L\Delta}
$$

$$
+ \left| 1 - \frac{\beta}{\alpha + \left\| \nabla f_i(x^k) - g_i^k \right\|} \right| \left\| \nabla f_i(x^k) - g_i^k \right\|.
$$

If $\left\| \nabla f_i(x^k) - g_i^k \right\| \leq R$ for all $i$, and $\frac{\beta}{\alpha + R} < 1$, then $\left\| \nabla f_i(x^{k+1}) - g_i^{k+1} \right\| \leq R$ when

$$
\gamma \leq \frac{\beta R}{\alpha + R} \frac{1}{R + \sqrt{2L\Delta}} \frac{1}{L_{\max}}.
$$

$\square$

From Lemma 3, we can establish the convergence theorem for $\alpha$-NormEC without server-side normalization in the non-private setting, similar to the one for $\alpha$-NormEC with server-side normalization in Theorem 1.

**Theorem 3.** *Consider $\alpha$-NormEC (Algorithm 1) without server normalization for solving Problem (1), where Assumptions 1 and 2 hold. Let $\beta, \alpha, \gamma > 0$ be chosen such that*

$$
\frac{\beta}{\alpha + R} < 1, \quad \text{and} \quad \gamma \leq \frac{\beta R}{\alpha + R} \frac{1}{R + \sqrt{2L\Delta}} \frac{1}{L_{\max}},
$$

*where $R = \max_{i \in [1,n]} \left\| \nabla f_i(x^0) - g_i^0 \right\|$ and $L_{\max} = \max_{i \in [1,n]} L_i$. Then,*

$$
\min_{k \in [0,K]} \left\| \nabla f(x^k) \right\| \leq \frac{f(x^0) - f^{\inf}}{\gamma(K+1)} + 2R + \frac{L}{2}\gamma.
$$

*Proof.* We prove the result in Theorem 3 in four steps.

**Step 1) Prove by induction that** $\left\| \nabla f_i(x^k) - g_i^k \right\| \leq R$ **for** $R = \max_{i \in [1,n]} \left\| \nabla f_i(x^0) - g_i^0 \right\|$. For $k = 0$, this is obvious. Next, let $\left\| \nabla f_i(x^l) - g_i^l \right\| \leq R$ for $R = \max_{i \in [1,n]} \left\| \nabla f_i(x^0) - g_i^0 \right\|$ for $l = 0, 1, \ldots, k$. Then, if $\beta/(\alpha + R) < 1$, and $\gamma \leq \frac{\beta R}{\alpha + R} \frac{1}{R + \sqrt{2L\Delta}} \frac{1}{L_{\max}}$, then from Lemma 3 $\left\| \nabla f_i(x^{k+1}) - g_i^{k+1} \right\| \leq R$.

**Step 2) Bound** $\left\| \nabla f_i(x^k) - g_i^{k+1} \right\|$. From the definition of the Euclidean norm,

$$
\left\| \nabla f_i(x^k) - g_i^{k+1} \right\| \overset{g_i^{k+1}}{=} \left\| \nabla f_i(x^k) - g_i^k - \beta \mathrm{Norm}_\alpha \left( \nabla f_i(x^k) - g_i^k \right) \right\|
$$

$$
\overset{\text{Lemma 1}}{\leq} \left| 1 - \frac{\beta}{\alpha + \left\| \nabla f_i(x^k) - g_i^k \right\|} \right| \left\| \nabla f_i(x^k) - g_i^k \right\|
$$

$$
\overset{(*)}{\leq} \left( 1 - \frac{\beta}{\alpha + R} \right) R \leq R,
$$

where we reach $(*)$ by the fact that $\left\| \nabla f_i(x^k) - g_i^k \right\| \leq R$, $\frac{\beta}{\alpha + R} < 1$, and $\gamma \leq \frac{\beta R}{\alpha + R} \frac{1}{R + \sqrt{2L\Delta}} \frac{1}{L_{\max}}$.

Finally, we follow Step 3) and Step 4) by following the proof arguments in Theorem 1 to obtain the final result.

$\square$

## B.6 EXTENSION TO STOCHASTIC GRADIENTS

We can leverage our analysis to study $\alpha$-NormEC, where clients compute their local stochastic gradients $\nabla f_i(x^k; \xi_{i,[b]}^k) = \frac{1}{b} \sum_{j=1}^{b} \nabla f_i(x^k; \xi_{i,j}^k)$ with the mini-batch size $b$, instead of full local gradients $\nabla f_i(x^k)$, in Step 4 of Algorithm 1.

We make the following assumption on the clients' local stochastic gradients, which is stronger than the unbiased and variance-bounded conditions of local stochastic gradients.

**Assumption 3.** *Let $\frac{1}{b} \sum_{j=1}^{b} \nabla f_i(x; \xi_{i,j})$ be the mini-batch stochastic gradient estimator of the local gradient $\nabla f_i(x)$ at client $i$ such that almost surely for some $\sigma > 0$ and for any $b > 0$,*

$$\left\| \frac{1}{b} \sum_{j=1}^{b} \nabla f_i(x; \xi_{i,j}) - \nabla f_i(x) \right\| \leq \frac{\sigma}{b}.$$

*Here, $\xi_{i,1}, \ldots, \xi_{i,b}$ are independent and identically distributed random variables drawn from the data distribution $\mathcal{D}_i$ at client $i$.*

From Assumption 3, we can bound the gradient error norm of $\alpha$-NormEC for stochastic optimization in the following lemma.

**Lemma 4.** *Consider $\alpha$-NormEC (Algorithm 1) that uses stochastic gradients $\nabla f_i(x^k; \xi_{i,[b]}^k) = \frac{1}{b} \sum_{j=1}^{b} \nabla f_i(x^k; \xi_{i,j}^k)$ for solving Problem (1), where Assumptions 1 and 3 hold. If $\left\| \nabla f_i(x^k; \xi_{i,[b]}^k) - g_i^k \right\| \leq R$, $\frac{\beta}{\alpha+R} < 1$, $b \geq \frac{4\sigma(\alpha+R)}{R}$, and $\gamma \leq \frac{1}{L_{\max}} \frac{\beta R}{2(\alpha+R)}$, then $\left\| \nabla f_i(x^{k+1}; \xi_i^{k+1}) - g_i^{k+1} \right\| \leq R.$*

*Proof.* From the definition of the Euclidean norm,

$$\left\| \nabla f_i(x^{k+1}; \xi_{i,[b]}^{k+1}) - g_i^{k+1} \right\| \overset{(12)}{\leq} \left\| \nabla f_i(x^{k+1}; \xi_{i,[b]}^{k+1}) - \nabla f_i(x^{k+1}) \right\|$$
$$+ \left\| \nabla f_i(x^k; \xi_{i,[b]}^k) - \nabla f_i(x^k) \right\|$$
$$+ \left\| \nabla f_i(x^{k+1}) - \nabla f_i(x^k) \right\|$$
$$+ \left\| \nabla f_i(x^k; \xi_{i,[b]}^k) - g_i^{k+1} \right\|$$

$$\overset{g_i^{k+1}}{\leq} \left\| \nabla f_i(x^{k+1}; \xi_{i,[b]}^{k+1}) - \nabla f_i(x^{k+1}) \right\|$$
$$+ \left\| \nabla f_i(x^k; \xi_{i,[b]}^k) - \nabla f_i(x^k) \right\|$$
$$+ \left\| \nabla f_i(x^{k+1}) - \nabla f_i(x^k) \right\|$$
$$+ \left\| \nabla f_i(x^k; \xi_{i,[b]}^k) - g_i^k - \beta \mathrm{Norm}_\alpha \left( \nabla f_i(x^k; \xi_{i,[b]}^k) - g_i^k \right) \right\|.$$

From Assumption 3,

$$\left\| \nabla f_i(x^{k+1}; \xi_{i,[b]}^{k+1}) - g_i^{k+1} \right\| \leq \frac{2\sigma}{b} + \left\| \nabla f_i(x^{k+1}) - \nabla f_i(x^k) \right\|$$
$$+ \left| 1 - \frac{\beta}{\alpha + \left\| \nabla f_i(x^k; \xi_{i,[b]}^k) - g_i^k \right\|} \right| \left\| \nabla f_i(x^k; \xi_{i,[b]}^k) - g_i^k \right\|$$

$$\overset{x^{k+1} + \text{Assumption 1}}{\leq} \frac{2\sigma}{b} + L_i \gamma$$
$$+ \left| 1 - \frac{\beta}{\alpha + \left\| \nabla f_i(x^k; \xi_{i,[b]}^k) - g_i^k \right\|} \right| \left\| \nabla f_i(x^k; \xi_{i,[b]}^k) - g_i^k \right\|.$$

If $\beta/(\alpha + R) < 1$, $b \geq \frac{4\sigma(\alpha+R)}{R}$, and $\gamma \leq \frac{1}{L_{\max}}\frac{\beta R}{2(\alpha+R)}$, then we can prove that $\left\|\nabla f_i(x^{k+1}; \xi_i^{k+1}) - g_i^{k+1}\right\| \leq R$. $\qquad\square$

From Lemma 4, we can establish the convergence theorem for $\alpha$-NormEC that uses local stochastic gradients.

**Theorem 4.** *Consider $\alpha$-NormEC (Algorithm 1) that uses stochastic gradients $\nabla f_i(x^k; \xi_{i,[b]}^k) = \frac{1}{b}\sum_{j=1}^b \nabla f_i(x^k; \xi_{i,j}^k)$ for solving Problem (1), where Assumptions 1 and 3 hold. Let $\beta, \alpha, \gamma > 0$ be chosen such that*

$$b \geq \frac{4\sigma(\alpha + R)}{R}, \quad \frac{\beta}{\alpha + R} < 1 \quad and \quad \gamma \leq \frac{\beta R}{2(\alpha + R)}\frac{1}{L_{\max}},$$

*where $R = \max_{i\in[1,n]}\left\|\nabla f_i(x^0; \xi_{i,[b]}^0) - g_i^0\right\|$ and $L_{\max} = \max_{i\in[1,n]} L_i$. Then, almost surely,*

$$\min_{k\in[0,K]}\left\|\nabla f(x^k)\right\| \leq \frac{f(x^0) - f^{\inf}}{\gamma(K+1)} + 2\left(\frac{\sigma}{b} + R\right) + \frac{L}{2}\gamma.$$

*Proof.* We prove the result in Theorem 3 in four steps.

**Step 1) Prove by induction that** $\left\|\nabla f_i(x^k; \xi_{i,[b]}^k) - g_i^k\right\| \leq R$ **for** $R = \max_{i\in[1,n]}\left\|\nabla f_i(x^0; \xi_{i,[b]}^0) - g_i^0\right\|$. For $k=0$, this is obvious. Next, let $\left\|\nabla f_i(x^l; \xi_{i,[b]}^l) - g_i^l\right\| \leq R$ for $R = \max_{i\in[1,n]}\left\|\nabla f_i(x^0; \xi_{i,[b]}^0) - g_i^0\right\|$ for $l = 0, 1, \ldots, k$. Then, from Lemma 4 $\left\|\nabla f_i(x^{k+1}; \xi_{i,[b]}^{k+1}) - g_i^{k+1}\right\| \leq R$.

**Step 2) Bound** $\left\|\nabla f_i(x^k) - g_i^{k+1}\right\|$. From the definition of the Euclidean norm,

$$
\begin{aligned}
\left\|\nabla f_i(x^k) - g_i^{k+1}\right\| \overset{g_i^{k+1}}{=} \quad & \left\|\nabla f_i(x^k) - g_i^k - \beta \mathrm{Norm}_\alpha\left(\nabla f_i(x^k; \xi_{i,[b]}^k) - g_i^k\right)\right\| \\
\overset{(12)}{\leq} \quad & \left\|\nabla f_i(x^k; \xi_{i,[b]}^k) - \nabla f_i(x^k)\right\| \\
& + \left\|\nabla f_i(x^k; \xi_{i,[b]}^k) - g_i^k - \beta \mathrm{Norm}_\alpha\left(\nabla f_i(x^k; \xi_{i,[b]}^k) - g_i^k\right)\right\| \\
\overset{\text{Lemma 1}}{\leq} \quad & \left\|\nabla f_i(x^k; \xi_{i,[b]}^k) - \nabla f_i(x^k)\right\| \\
& + \left|1 - \frac{\beta}{\alpha + \left\|\nabla f_i(x^k; \xi_{i,[b]}^k) - g_i^k\right\|}\right|\left\|\nabla f_i(x^k; \xi_{i,[b]}^k) - g_i^k\right\|.
\end{aligned}
$$

From Assumption 3,

$$\left\|\nabla f_i(x^k) - g_i^{k+1}\right\| \leq \frac{\sigma}{b} + \left|1 - \frac{\beta}{\alpha + \left\|\nabla f_i(x^k; \xi_{i,[b]}^k) - g_i^k\right\|}\right|\left\|\nabla f_i(x^k; \xi_{i,[b]}^k) - g_i^k\right\|.$$

By the fact that $\left\|\nabla f_i(x^k; \xi_{i,[b]}^k) - g_i^k\right\| \leq R$, and $\beta/(\alpha + R) < 1$ from Step 1) of the proof,

$$\left\|\nabla f_i(x^k) - g_i^{k+1}\right\| \leq \frac{\sigma}{b} + \left(1 - \frac{\beta}{\alpha + R}\right)R \leq \frac{\sigma}{b} + R.$$

Finally, we follow Step 3) and Step 4) by following the proof arguments in Theorem 1 to obtain the final result. $\qquad\square$

Theorem 4 states that $\alpha$-NormEC using local stochastic gradients achieves the sublinear, almost sure convergence up to the additive constants $2(\sigma/b + R) + \gamma L/2$. Here, the mini-batch size $b$ must be sufficiently large to ensure the convergence and to improve the accuracy of the solution obtained from running the method.

## C  PRIVATE RESULTS

### C.1  PROOF OF THEOREM 2

We prove Theorem 2 by two useful lemmas:

1. Lemma 2, which states $\left\|\nabla f_i(x^{k+1}) - g_i^{k+1}\right\| \leq R$ for some positive scalars $R$, given that $\left\|\nabla f_i(x^k) - g_i^k\right\| \leq R$ and the hyperparameters $\gamma, \beta, \alpha$ are properly tuned, and

2. Lemma 5, which bounds the difference in expectation between the memory vectors maintained by the central server and clients.

**Lemma 5** (DP setting). *Consider* DP-$\alpha$-NormEC *(Algorithm 1) for solving Problem (1), where Assumption 1 holds. If $\hat{g}^0 = \frac{1}{n} \sum_{i=1}^n g_i^0$, then*

$$\mathrm{E}\left[\left\|\hat{g}^{k+1} - \frac{1}{n} \sum_{i=1}^n g_i^{k+1}\right\|\right] \leq \sqrt{\frac{\beta^2(K+1)\sigma_{\mathrm{DP}}^2}{n}}.$$

*Proof.* From the definition of $g_i^k$ and $\hat{g}^k$,

$$e^{k+1} = e^k + \beta z^{k+1},$$

where $e^k = \hat{g}^k - \frac{1}{n} \sum_{i=1}^n g_i^k$, and $z^k = \frac{1}{n} \sum_{i=1}^n z_i^k$. By applying the equation recursively,

$$e^{k+1} = e^0 + \beta \sum_{l=1}^{k+1} z^l.$$

Therefore, by the triangle inequality,

$$\left\|e^{k+1}\right\| \leq \left\|e^0\right\| + \left\|\beta \sum_{l=1}^{k+1} z^l\right\|.$$

If $\hat{g}^0 = \frac{1}{n} \sum_{i=1}^n g_i^0$, then $e^0 = 0$ and therefore

$$\left\|e^{k+1}\right\| \leq \left\|\beta \sum_{l=1}^{k+1} z^l\right\|.$$

By taking the expectation,

$$\mathrm{E}\left[\left\|e^{k+1}\right\|\right] \leq \mathrm{E}\left[\left\|\beta \sum_{l=1}^{k+1} z^l\right\|\right]$$

$$= \mathrm{E}\left[\sqrt{\left\|\beta \sum_{l=1}^{k+1} z^l\right\|^2}\right]$$

$$\leq \sqrt{\mathrm{E}\left[\left\|\beta \sum_{l=1}^{k+1} z^l\right\|^2\right]},$$

where we reach the last inequality by Jensen's inequality. Next, by expanding the terms,

$$\mathrm{E}\left[\left\|e^{k+1}\right\|\right] \leq \sqrt{\beta^2 \sum_{l=1}^{k+1} \mathrm{E}\left[\|z^l\|^2\right] + \beta^2 \sum_{j \neq i} \mathrm{E}\left[\langle z^i, z^j \rangle\right]}$$

$$\overset{(*)}{=} \sqrt{\beta^2 \sum_{l=1}^{k+1} \mathrm{E}\left[\|z^l\|^2\right]}$$

$$\overset{(\ddagger)}{\leq} \sqrt{\frac{\beta^2}{n} \sum_{l=1}^{k+1} \sigma_{\mathrm{DP}}^2},$$

where we reach $(*)$ by the fact that $\mathrm{E}\left[\langle z^j, z^i\rangle\right] = 0$ for $i \neq j$, and $(\ddagger)$ by the fact that $\mathrm{E}\left[\left\|z^k\right\|^2\right] = \sigma_{\mathrm{DP}}^2/n$ (as $z_i^k$ is independent of $z_j^k$ for $i \neq j$). Therefore,

$$
\begin{aligned}
\mathrm{E}\left[\left\|e^{k+1}\right\|\right] &\leq \sqrt{\frac{\beta^2(k+1)\sigma_{\mathrm{DP}}^2}{n}} \\
&\overset{k \leq K}{\leq} \sqrt{\frac{\beta^2(K+1)\sigma_{\mathrm{DP}}^2}{n}}.
\end{aligned}
$$

$\square$

Now, we prove Theorem 2 in three steps.

**Step 1) Prove by induction that** $\left\|\nabla f_i(x^k) - g_i^k\right\| \leq R$ **for** $R = \max_{i \in [1,n]} \left\|\nabla f_i(x^0) - g_i^0\right\|$. For $k = 0$, this is obvious. Next, let $\left\|\nabla f_i(x^l) - g_i^l\right\| \leq R$ for $R = \max_{i \in [1,n]} \left\|\nabla f_i(x^0) - g_i^0\right\|$ for $l = 0, 1, \ldots, k$. Then, if $\beta/(\alpha + R) < 1$, and $\gamma \leq \frac{\beta R}{\alpha + R}\frac{1}{L_{\max}}$, then from Lemma 2 $\left\|\nabla f_i(x^{k+1}) - g_i^{k+1}\right\| \leq R$.

**Step 2) Bound** $\left\|\nabla f_i(x^k) - g_i^{k+1}\right\|$. From the definition of the Euclidean norm,

$$
\begin{aligned}
\left\|\nabla f_i(x^k) - g_i^{k+1}\right\| &\overset{g_i^{k+1}}{=} \left\|\nabla f_i(x^k) - g_i^k - \beta \mathrm{Norm}_\alpha\left(\nabla f_i(x^k) - g_i^k\right)\right\| \\
&\overset{\text{Lemma 2}}{\leq} \left|1 - \frac{\beta}{\alpha + \left\|\nabla f_i(x^k) - g_i^k\right\|}\right| \left\|\nabla f_i(x^k) - g_i^k\right\|.
\end{aligned}
$$

**Step 3) Derive the descent inequality in** $\mathrm{E}\left[f(x^k) - f^{\inf}\right]$. Denote $g^k = \frac{1}{n}\sum_{i=1}^n g_i^k$. By the $L$-smoothness of $f$, and by the definition of $x^{k+1}$,

$$
\begin{aligned}
f(x^{k+1}) - f^{\inf} &\leq f(x^k) - f^{\inf} - \frac{\gamma}{\|\hat{g}^{k+1}\|}\left\langle\nabla f(x^k), \hat{g}^{k+1}\right\rangle + \frac{L\gamma^2}{2} \\
&= f(x^k) - f^{\inf} - \gamma\left\|\hat{g}^{k+1}\right\| + \frac{\gamma}{\|\hat{g}^{k+1}\|}\left\langle\nabla f(x^k) - \hat{g}^{k+1}, \hat{g}^{k+1}\right\rangle + \frac{L\gamma^2}{2} \\
&\overset{(11)}{\leq} f(x^k) - f^{\inf} - \gamma\left\|\hat{g}^{k+1}\right\| + \gamma\left\|\nabla f(x^k) - \hat{g}^{k+1}\right\| + \frac{L\gamma^2}{2} \\
&\overset{(12)}{\leq} f(x^k) - f^{\inf} - \gamma\left\|\nabla f(x^k)\right\| + 2\gamma\left\|\nabla f(x^k) - \hat{g}^{k+1}\right\| + \frac{L\gamma^2}{2} \\
&\overset{(13)}{\leq} f(x^k) - f^{\inf} - \gamma\left\|\nabla f(x^k)\right\| + 2\gamma\frac{1}{n}\sum_{i=1}^n\left\|\nabla f_i(x^k) - g_i^{k+1}\right\| \\
&\quad + 2\gamma\left\|\hat{g}^{k+1} - g^{k+1}\right\| + \frac{L\gamma^2}{2}.
\end{aligned}
$$

Next, let $\mathcal{F}^k$ be the history up to iteration $k$, i.e. $\mathcal{F}^k := \{x^0, z_1^0, \ldots, z_n^0, \ldots, x^k, z_1^k, \ldots, z_n^k\}$. Then,

$$
\begin{aligned}
\mathrm{E}\left[f(x^{k+1}) - f^{\inf}\big|\mathcal{F}^k\right] &\leq f(x^k) - f^{\inf} - \gamma\left\|\nabla f(x^k)\right\| + 2\gamma\frac{1}{n}\sum_{i=1}^n\mathrm{E}\left[\left\|\nabla f_i(x^k) - g_i^{k+1}\right\|\big|\mathcal{F}^k\right] \\
&\quad + 2\gamma\mathrm{E}\left[\left\|\hat{g}^{k+1} - g^{k+1}\right\|\big|\mathcal{F}^k\right] + \frac{L\gamma^2}{2}.
\end{aligned}
$$

Next, by the upper-bound for $\left\|\nabla f_i(x^k) - g_i^{k+1}\right\|$,

$$
\begin{aligned}
\mathrm{E}\left[\left\|\nabla f_i(x^k) - g_i^{k+1}\right\|\big|\mathcal{F}^k\right] &\leq \mathrm{E}\left[\left|1 - \frac{\beta}{\alpha + \left\|\nabla f_i(x^k) - g_i^k\right\|}\right|\left\|\nabla f_i(x^k) - g_i^k\right\|\Big|\mathcal{F}^k\right] \\
&= \left|1 - \frac{\beta}{\alpha + \left\|\nabla f_i(x^k) - g_i^k\right\|}\right|\left\|\nabla f_i(x^k) - g_i^k\right\| \\
&\leq \left(1 - \frac{\beta}{\alpha + R}\right)R \leq R,
\end{aligned}
$$

where we reach the second inequality by the fact that $\left\|\nabla f_i(x^k) - g_i^k\right\| \leq R$, $\frac{\beta}{\alpha+R} < 1$, and $\gamma \leq \frac{\beta R}{\alpha+R}\frac{1}{L_{\max}}$. Thus,

$$
\begin{aligned}
\mathrm{E}\left[f(x^{k+1}) - f^{\inf}\middle|\mathcal{F}^k\right] &\leq f(x^k) - f^{\inf} - \gamma\left\|\nabla f(x^k)\right\| + 2\gamma R \\
&\quad + 2\gamma\mathrm{E}\left[\left\|\hat{g}^{k+1} - g^{k+1}\right\|\middle|\mathcal{F}^k\right] + \frac{L\gamma^2}{2}.
\end{aligned}
$$

By taking the expectation, and by the tower property $\mathrm{E}\left[\mathrm{E}\left[X\middle|Y\right]\right] = \mathrm{E}\left[X\right]$,

$$
\begin{aligned}
\mathrm{E}\left[f(x^{k+1}) - f^{\inf}\right] &= \mathrm{E}\left[\mathrm{E}\left[f(x^{k+1}) - f^{\inf}\middle|\mathcal{F}^k\right]\right] \\
&\leq \mathrm{E}\left[f(x^k) - f^{\inf}\right] - \gamma\mathrm{E}\left[\left\|\nabla f(x^k)\right\|\right] + 2\gamma R \\
&\quad + 2\gamma\mathrm{E}\left[\left\|\hat{g}^{k+1} - g^{k+1}\right\|\right] + \frac{L\gamma^2}{2}.
\end{aligned}
$$

Next, by using Lemma 5,

$$
\begin{aligned}
\mathrm{E}\left[f(x^{k+1}) - f^{\inf}\right] &\leq \mathrm{E}\left[f(x^k) - f^{\inf}\right] - \gamma\mathrm{E}\left[\left\|\nabla f(x^k)\right\|\right] + 2\gamma R \\
&\quad + 2\gamma\sqrt{\frac{\beta^2(K+1)\sigma_{\mathrm{DP}}^2}{n}} + \frac{L\gamma^2}{2}.
\end{aligned}
$$

Therefore,

$$
\begin{aligned}
\min_{k\in[0,K]}\mathrm{E}\left[\left\|\nabla f(x^k)\right\|\right] &\leq \frac{1}{K+1}\sum_{k=0}^{K}\mathrm{E}\left[\left\|\nabla f(x^k)\right\|\right] \\
&\leq \frac{\mathrm{E}\left[f(x^0) - f^{\inf}\right] - \mathrm{E}\left[f(x^{K+1}) - f^{\inf}\right]}{\gamma(K+1)} + 2R + 2\sqrt{\frac{\beta^2(K+1)\sigma_{\mathrm{DP}}^2}{n}} + \frac{L}{2}\gamma \\
&\leq \frac{f(x^0) - f^{\inf}}{\gamma(K+1)} + 2R + 2\sqrt{\frac{\beta^2(K+1)\sigma_{\mathrm{DP}}^2}{n}} + \frac{L}{2}\gamma,
\end{aligned}
$$

where we reach the last inequality by the fact that $f^{\inf} \geq f(x^{K+1})$.

### C.1.1 DISCUSSION

By choosing $g_i^0$ such that $R = \frac{D}{(K+1)^{1/6}}$ with any $D > 0$, $\beta = \frac{\beta_0}{(K+1)^{2/3}}$ with $\beta_0 \in (0,1]$, $\alpha > 1$, and $\gamma \leq \frac{A}{(K+1)^{5/6}}$ with $A = \frac{\beta_0 D}{L_{\max}(\alpha+D)}$, then the conditions for $\beta, \alpha, \gamma$ in Theorem 2 are satisfied, and from Theorem 2 DP-$\alpha$-NormEC attains the $\mathcal{O}(1/K^{1/6})$ convergence rate in the gradient norm:

$$
\min_{k\in[0,K]}\mathrm{E}\left[\left\|\nabla f(x^k)\right\|\right] \leq \frac{C}{(K+1)^{1/6}} + \frac{LA}{2(K+1)^{5/6}},
$$

where $C_1 = \frac{f(x^0) - f^{\inf}}{A} + 2D + 2\beta_0\sigma_{\mathrm{DP}}$.

### C.2 UTILITY GUARANTEE OF DP-$\alpha$-NormEC

In this section, we present the utility guarantee of DP-$\alpha$-NormEC.

**Corollary 2** (Utility guarantee in DP setting). *Consider* DP-$\alpha$-NormEC *(Algorithm 1) for solving Problem (1) under the same setting as Theorem 2. If* $\sigma_{\mathrm{DP}} = \mathcal{O}(\sqrt{(K+1)\log(1/\delta)}\epsilon^{-1})$, $\beta = \frac{\beta_0}{K+1}$ *with* $\beta_0 = \mathcal{O}\left(\sqrt{\frac{\Delta}{A}}\right)$ *and* $\alpha = R = \mathcal{O}\left(\sqrt[4]{d}\sqrt{\Delta A}\right)$ *for* $\Delta = \sqrt{L_{\max}(f(x^0) - f^{\inf})}$ *and* $A = \frac{\sqrt{\log(1/\delta)}}{\sqrt{n}\epsilon}$*, then Algorithm 1 satisfies* $(\epsilon, \delta)$*-DP and attains the bound*

$$
\min_{k\in[0,K]}\mathrm{E}\left[\left\|\nabla f(x^k)\right\|\right] \leq \mathcal{O}\left(\Delta\sqrt[4]{d\frac{\log(1/\delta)}{n\epsilon^2}}\right).
$$

**Proof:** Let $\sigma_{\text{DP}} = \mathcal{O}\left(\frac{\sqrt{(K+1)\log(1/\delta)}}{\epsilon}\right)$, and $\beta = \frac{\beta_0}{K+1}$ with $0 < \beta_0 < \alpha + R$. Then, from Theorem 2, we get $\gamma \le \frac{\beta_0 R}{\alpha + R} \frac{1}{L_{\max}} \frac{1}{K+1}$ with $R = \max_{i \in [1,n]} \left\| \nabla f_i(x^0) - g_i^0 \right\|$, and

$$
\begin{aligned}
\min_{k \in [0,K]} \mathrm{E}\left[\left\| \nabla f(x^k) \right\|\right] &\le \frac{L_{\max}(\alpha + R)(f(x^0) - f^{\inf})}{\beta_0 R} + 2R + 2\frac{\beta_0 \sqrt{\log(1/\delta)}}{\sqrt{n}\epsilon} \\
&\quad + \frac{L\beta_0 R}{2(\alpha + R)L_{\max}} \frac{1}{K+1}.
\end{aligned}
$$

If $\beta_0 = \mathcal{O}\left(\sqrt{\frac{L_{\max}(f(x^0) - f^{\inf})}{A}}\right)$ and $\alpha = R = \mathcal{O}\left(\sqrt[4]{d}\sqrt{L_{\max}(f(x^0) - f^{\inf})A}\right)$ for $A = \frac{\sqrt{\log(1/\delta)}}{\sqrt{n}\epsilon}$, then

$$
\min_{k \in [0,K]} \mathrm{E}\left[\left\| \nabla f(x^k) \right\|\right] \le \mathcal{O}\left(\sqrt[4]{d}\sqrt{L_{\max}(f(x^0) - f^{\inf})A}\right).
$$

### C.3  PRIVATE INITIALIZATION OF THE MEMORY VECTORS

Secure aggregation can be used in $\alpha$-NormEC to initialize the server's aggregated memory vector, $\hat{g}^0 = \frac{1}{n}\sum_{i=1}^n g_i^0$, while keeping the clients' individual memory vectors, $g_i^0$, private. For example, clients that share a random seed can add and subtract identical cryptographic noise, $h$, from their respective vectors (e.g., $g_1^0 + h$ and $g_2^0 - h$). When these are aggregated by the server, the noise cancels out, preserving the accuracy of the average: $\frac{1}{2}(g_1^0 + h) + \frac{1}{2}(g_2^0 - h) = \frac{1}{2}(g_1^0 + g_2^0)$. This method protects the individual vectors from the server without compromising the accuracy of the overall average.

Furthermore, we can extend this initialization for $\hat{g}^0$ to include general DP noise. Specifically, we can set $\hat{g}^0 = \frac{1}{n}\sum_{i=1}^n (g_i^0 + e_i^0)$, where $e_i^0$ is the DP noise that Client $i$ adds to its local memory vector before communicating it to the server. Our analysis, especially in Lemma 5, shows that this generalized initialization can be accommodated. The worst-case bound for the error term, $\mathrm{E}\left[\left\| \hat{g}^{k+1} - \frac{1}{n}\sum_{i=1}^n g_i^{k+1} \right\|\right]$, will simply include an additional term, $e = \frac{1}{n}\sum_{i=1}^n e_i^0$, i.e.

$$
\mathrm{E}\left[\left\| \hat{g}^{k+1} - \frac{1}{n}\sum_{i=1}^n g_i^{k+1} \right\|\right] \le \sqrt{\frac{\beta^2(K+1)}{n}\sigma_{\text{DP}}^2} + \sqrt{\frac{1}{n}\sum_{i=1}^n \|e_i^0\|}.
$$

## D   EXPERIMENTAL DETAILS AND ADDITIONAL RESULTS

We include details on experimental setups and additional results in the non-private and private training for the ResNet20 model on the CIFAR-10 dataset.

### D.1   ADDITIONAL EXPERIMENTAL DETAILS

The dataset was split into train (90%) and test (10%) sets. The train samples were randomly shuffled and distributed across 10 workers. Every worker computed gradients with batch size 32. The training was performed for 300 communication rounds. The random seed was fixed to 42 for reproducibility.

All the methods were run with a constant step size (learning rate) without other techniques, such as schedulers, warm-up, or weight decay. They were evaluated across the following hyperparameter combinations:

- step size $\gamma$: $\{0.001, 0.01, 0.1, 1.0\}$,
- Sensitivity/clip threshold $\beta$: $\{0.01, 0.1, 1.0, 10.0\}$,
- Smoothed normalization value $\alpha$: $\{0.01, 0.1, 1.0\}$.

Our implementation is based on the public GitHub repository Idelbayev. Experiments were performed on a machine with a single GPU: NVIDIA GeForce RTX 3090.

### D.2   NON-PRIVATE TRAINING

#### D.2.1   SENSITIVITY OF $\alpha$-NormEC TO PARAMETERS $\beta, \alpha$

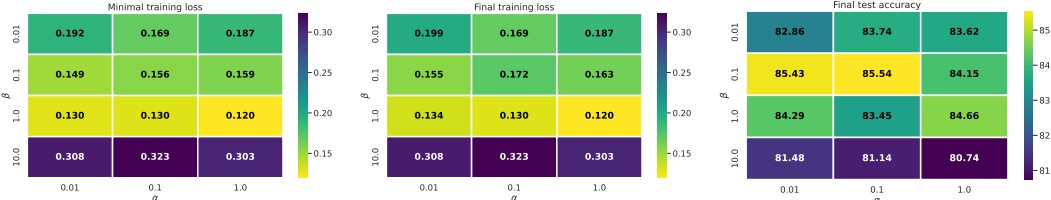

Figure 4: **Minimal** train loss (left), **final** train loss (middle), and **final** test accuracy (right) achieved by non-private $\alpha$-NormEC, after 300 communication rounds using a fine-tuned constant step size $\gamma$.

Figure 4 supplements the results in Figure 1 with other metrics. Figure 5 displays convergence curves across different combinations of $\alpha, \beta$ parameters with optimally selected step sizes $\gamma$.

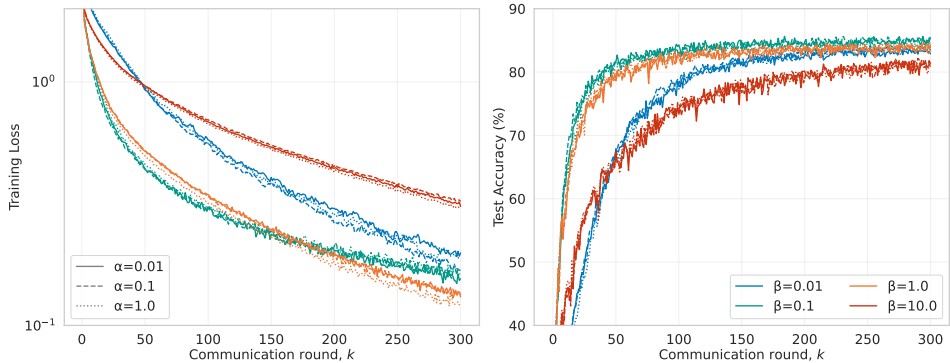

Figure 5: Training loss and test accuracy of non-private $\alpha$-NormEC with $\alpha = 0.01$ [solid], 0.1 [dashed], and 1.0 [dotted], and $\beta = 0.01$ [blue], 0.1 [green], 1.0 [orange], and 10.0 [red].

### D.2.2 BENEFITS OF ERROR COMPENSATION

Leveraging error compensation (EC), $\alpha$-NormEC without server normalization achieves superior performance compared to DP-SGD with direct smoothed normalization across a range of $\beta$ and $\gamma$ hyperparameter settings (where $\alpha = 0.01$), in terms of the final test accuracy reported in Figure 6 and Table 7. From Table 7, $\alpha$-NormEC without server normalization consistently outperforms DP-SGD across most combinations. This trend is particularly evident for small $\beta$ values ($\beta = 0.01$), where DP-SGD achieves only 51.10% accuracy while $\alpha$-NormEC reaches 84.04%. The only exception is $\beta = 10.0$, where DP-SGD outperforms $\alpha$-NormEC. However, this combination is less practical in the private setting, as too high $\beta$ values imply high private noise, thus leading to slow algorithmic convergence.

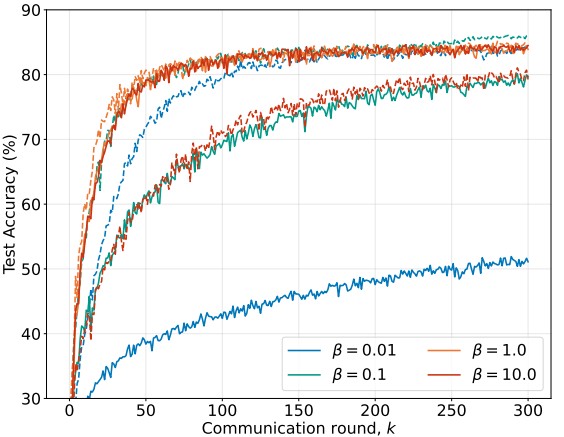

| Method | $\beta$ | $\gamma$ | Final Accuracy |
|---|---|---|---|
| $\alpha$-NormEC | 0.01 | 0.1 | 84.04% |
| | 0.1 | 0.1 | **86.09**% |
| | 1.0 | 0.1 | 84.80% |
| | 10.0 | 0.01 | 79.25% |
| DP-SGD (2) | 0.01 | 1.0 | 51.10% |
| | 0.1 | 1.0 | 79.68% |
| | 1.0 | 1.0 | 83.89% |
| | 10.0 | 0.1 | 84.50% |

Figure 7: Best configurations and final test accuracies.

Figure 6: Comparison of DP-SGD (2) [solid] and $\alpha$-NormEC (1) [dashed] without server normalization.

### D.2.3 EFFECT OF SERVER NORMALIZATION

We investigate the impact of server-side normalization (Line 11 in Algorithm 1) on the convergence performance of $\alpha$-NormEC. We reported training loss and test accuracy of $\alpha$-NormEC without and with server normalization in Figure 8 while summarizing their final test accuracy in Table 3.

$\alpha$-NormEC without server normalization generally achieves faster convergence in training loss and higher test accuracy than $\alpha$-NormEC with server normalization across varying $\beta$ values. Notably, at $\beta = 0.1$, $\alpha$-NormEC without server normalization achieves the highest test accuracy of **86.09%**. Only at the large value of $\beta = 10.0$ does server normalization improve the test accuracy of $\alpha$-NormEC without server normalization by approximately 2.2%.

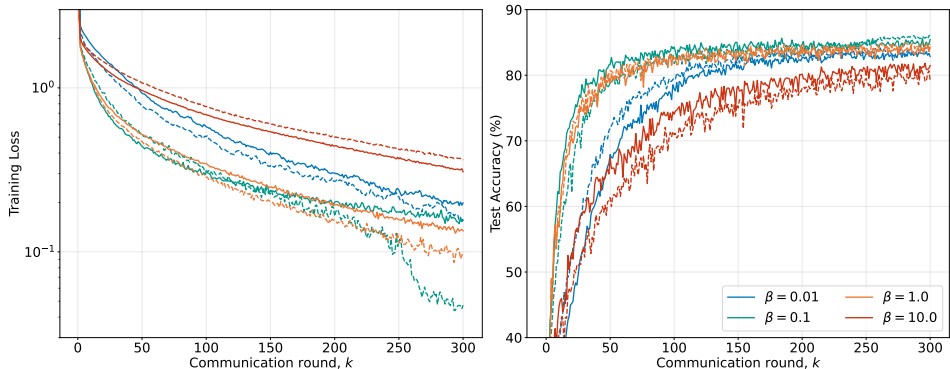

Figure 8: Training loss and test accuracy of $\alpha$-NormEC with [solid] and without [dashed] server normalization.

| Method: $\alpha$-NormEC | $\beta$ | $\gamma$ | Final Accuracy |
|---|---|---|---|
| With server normalization | 0.01 | 0.01 | 82.86% |
| | 0.1 | 0.1 | 85.43% |
| | 1.0 | 0.1 | 84.29% |
| | 10.0 | 0.1 | 81.48% |
| Without server normalization | 0.01 | 0.1 | 84.04% |
| | 0.1 | 0.1 | **86.09**% |
| | 1.0 | 0.1 | 84.80% |
| | 10.0 | 0.01 | 79.25% |

Table 3: Best configurations and final test accuracies of $\alpha$-NormEC with and without server normalization.

### D.2.4 COMPARISON OF Clip21 AND $\alpha$-NormEC

Figure 9 and Table 10 show that $\alpha$-NormEC without server normalization[4] achieves comparable convergence performance to Clip21 for most $\beta$ values. At small $\beta$ values $(0.01, 0.1)$, $\alpha$-NormEC without server normalization attains slightly lower final test accuracy. However, at high $\beta = 10.0$, Clip21 maintains the higher test accuracy, as the large clipping threshold effectively disables clipping. Furthermore, in most cases, both methods achieve their best performance with $\gamma = 0.1$, except for $\alpha$-NormEC at $\beta = 10.0$, where a smaller learning rate $(\gamma = 0.01)$ was optimal.

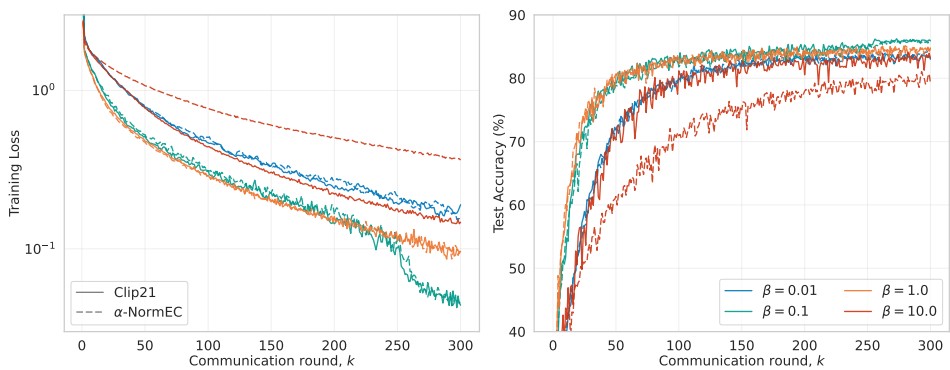

Figure 9: Training loss and test accuracy of Clip21 [solid] and $\alpha$-NormEC [dashed] without server normalization in the non-private training.

| Method | $\beta$ | $\gamma$ | Final Accuracy |
|---|---|---|---|
| Clip21 | 0.01 | 0.1 | 83.00% |
| | 0.1 | 0.1 | 85.91% |
| | 1.0 | 0.1 | 84.78% |
| | 10.0 | 0.1 | 83.19% |
| $\alpha$-NormEC | 0.01 | 0.1 | 84.04% |
| | 0.1 | 0.1 | **86.09**% |
| | 1.0 | 0.1 | 84.80% |
| | 10.0 | 0.01 | 79.25% |

Figure 10: Best configurations and final test accuracies.

---

[4]We ran $\alpha$-NormEC without server normalization because it showed better performance than $\alpha$-NormEC with server normalization according to Appendix D.2.3.

### D.3 PRIVATE TRAINING

We complement the results in Section 5.2 with test accuracy convergence curves in Figure 13 (right). Additionally, Figures 12 and 11 present a comprehensive heatmap analysis of the highest test accuracy achieved by DP-$\alpha$-NormEC with and without server normalization (SN) and Clip21 across different privacy levels ($\beta$) and learning rates ($\gamma$).

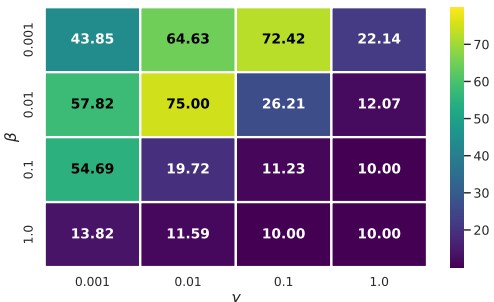

Figure 11: The highest test accuracy of DP-Clip21.

The heatmaps reveal that without server normalization, performance is highly sensitive to hyperparameter selection, with accuracy ranging from 10% to 77.56% depending on the specific $\beta$-$\gamma$ combination. With server normalization, this sensitivity is significantly reduced, with performance varying more gradually across the parameter space. The rightmost heatmap quantifies this difference, showing that server normalization provides substantial benefits (up to +53.49%) at high privacy levels ($\beta = 1.0$) and larger step sizes, while the non-normalized version can perform better at lower privacy levels with specific step sizes.

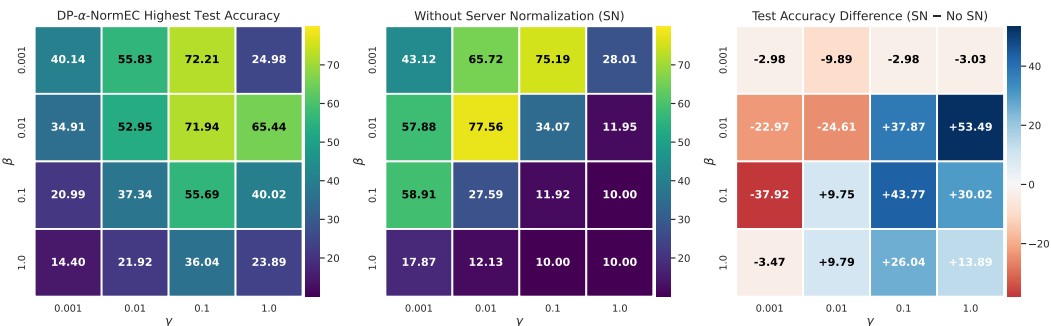

Figure 12: The highest test accuracy of DP-$\alpha$-NormEC with [left] and without [center] Server Normalization (SN), and their difference [right].

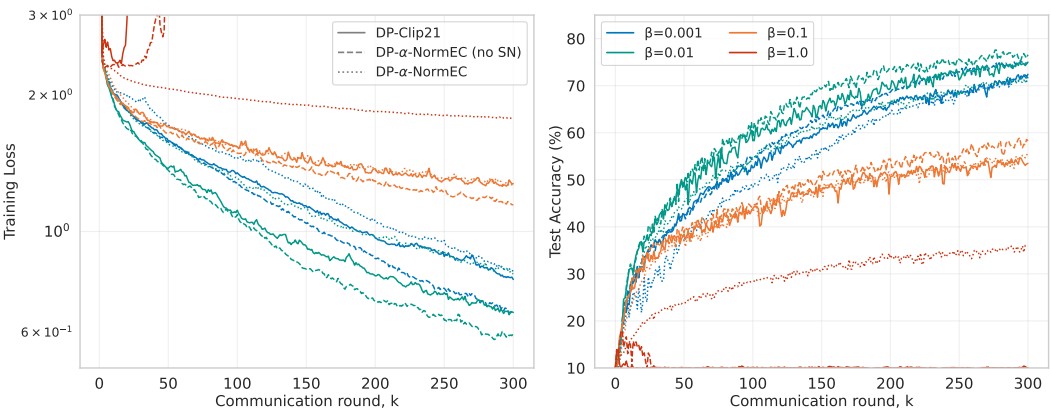

Figure 13: Training loss and test accuracy of DP-Clip21 [solid], and DP-$\alpha$-NormEC with [dotted] and without [dashed] server normalization (SN) across different $\beta$ values (with fine-tuned step sizes).

### D.3.1 STRICTER PRIVACY BUDGET ($\epsilon = 1$)

To validate our method under a stricter privacy guarantee, we present additional results for $\epsilon = 1$ in Figure 14. As expected, the increased DP noise in this high-privacy regime reduces the overall performance for both methods compared to the $\epsilon = 8$ setting. However, the results demonstrate a consistent performance advantage and smaller variability for DP-$\alpha$-NormEC over DP-Clip21. For

each corresponding hyperparameter setting for $\beta$, DP-$\alpha$-NormEC achieves both a lower training loss and higher test accuracy, confirming its effectiveness in this practical setting.

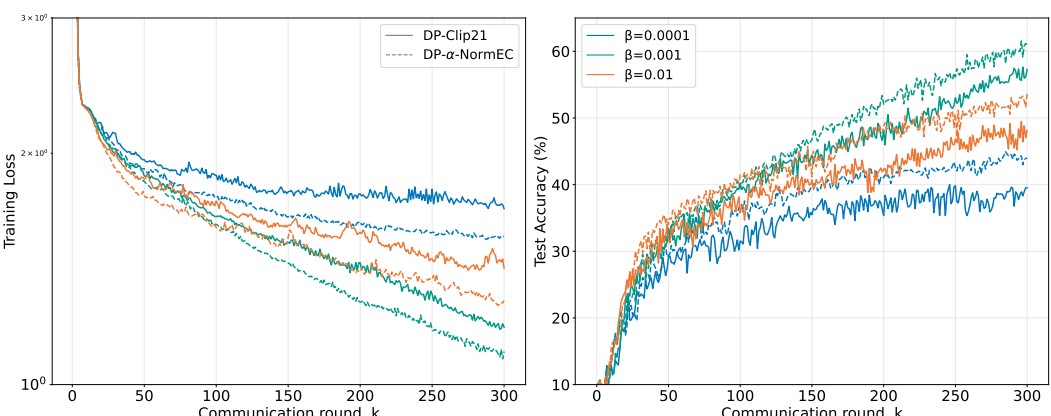

Figure 14: Training loss and test accuracy of DP-Clip21 [solid] and DP-$\alpha$-NormEC [dashed] across different $\beta$ values for $\epsilon = 1$. DP-$\alpha$-NormEC demonstrates both faster convergence and higher final accuracy for each $\beta$.

### D.3.2 SHORTER TRAINING

We present additional results in Figures 15, 16 by running DP-$\alpha$-NormEC for **150 communication rounds**. The step size $\gamma$ is tuned for every parameter $\beta$. In the non-private setting, (reasonably) longer training is basically always beneficial. However, in the private scenario, it may not hold due to increased noise variance as it scales with a number of iterations. Interestingly, we observe that for $\beta = 1$, the highest achieved accuracy after 150 iterations is almost the same as after a doubled communication budget of 300 iterations.

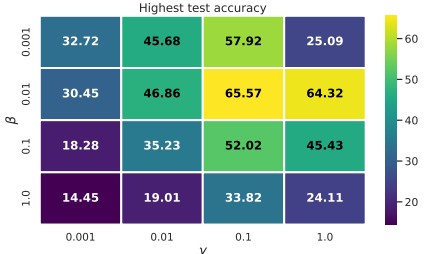

Figure 15: Best test accuracy of DP-$\alpha$-NormEC across different $\beta, \gamma$ values.

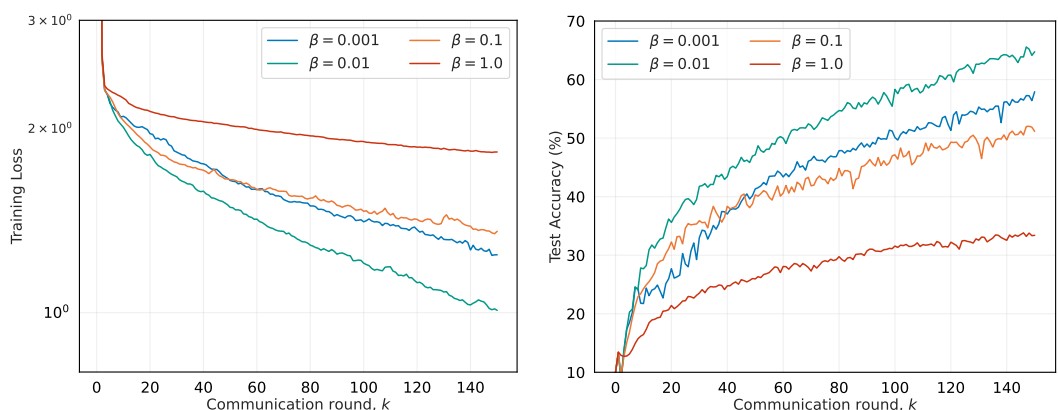

Figure 16: Training loss and test accuracy of DP-$\alpha$-NormEC across different $\beta$ values.

## ACKNOWLEDGMENTS

The authors used large language models (LLMs) during the preparation of this paper to assist with grammar, wording, and code implementation. No LLMs were used to write scientific content, or search for citations or related work. This is in accordance with two main LLM-related policies.

