# OpenReview forum: "Provably Convergent and Private Distributed Optimization via Smoothed Normalization"
_ICLR.cc/2026/Conference — Submitted to ICLR 2026_

### Official Review · Reviewer_Qzmf · 2025-10-24

**Soundness:** 3
**Presentation:** 3
**Contribution:** 3
**Rating:** 4
**Confidence:** 3

**Summary:**

This paper proposes $\alpha$-NormEC, which integrates smoothed normalization and error compensation for advanced federated learning (FL), and its private variant DP-$\alpha$-NormEC. While the algorithmic components are combinations of known techniques, the paper organizes several theoretical contributions. Under only the L-smooth assumption (i.e., without using bounded gradient norms or bounded data heterogeneity assumptions), it establishes for non-convex, smooth problems: (i) convergence of $\alpha$-NormEC, and (ii) provable convergence of DP-$\alpha$-NormEC in the private setting.

**Strengths:**

**(S1) Clear presentation**
The major contributions are clearly summarized in Section 1.1 and Table 1, and the exposition remains consistent throughout.

**(S2) Sufficient theoretical contributions**

(S2a) Convergence of gradient norms for L-smooth non-convex objectives using $\alpha$-NormEC (Theorem 1). The key appears to be Lemma 2 and an appropriate choice of parameters $(\alpha,\beta,\gamma)$, which together bound the norm of the difference between the full gradient and the stochastic gradient.

(S2b) Convergence of gradient norms for L-smooth non-convex objectives in the private setting using DP-$\alpha$-NormEC (Theorem 2). Given the private setting and the minimal assumptions, this is likely among the first provable convergence results of this setting.

**(S3) Sufficient discussion against baselines**This paper discusses how the obtained results (Theorems 1 and 2) compare with baselines (e.g., Clip21, EF21), which helps clarify the contribution of this paper.

**Weaknesses:**

**(W1) Are the convergence rates tight? (Theorem 1)**
Deriving convergence rates without any bounded heterogeneity assumption seems elegant; however, empirical FL behavior often depends strongly on data heterogeneity. Hence, the presented rates may not be tight in practice. A heterogeneity-aware convergence rate could plausibly be tighter.

**(W2) Concerns about initialization (Theorem 2)**
My understanding is that the key point in Theorem 2 is in the initialization (e.g., $g_i^0$ and parameters $\beta,\gamma$) to reduce the additive term introduced by the privacy setting. Since I believe this is important, it would be better to include a concise recipe in the main paper. I skimmed C.1.1, but it remained somewhat opaque. Could you provide an additional explanation?

**(W3) Limited experiments (Sec. 5)**
While I recognize this paper’s main contributions are on the theoretical side, the experimental section feels limited.

(W3a) Few benchmarks. The evaluation uses only CIFAR-10 with ResNet-20; can you validate on additional datasets/models?

(W3b) Baselines. It seems comparisons to strong SOTA methods are missing in both non-private and private settings (e.g., SCAFFOLD and DP-SCAFFOLD).

(W3c) Validation of data heterogeneity. Related to W1: I recommend that you demonstrate performance dependence on heterogeneity by controlling non-IIDness through the Dirichlet concentration parameter, which is a standard approach in FL.

**Questions:**

(Comment 1) This is just a suggestion: since the proposed method (Sec. 4) appears from p.6 onward, Sections 1–2 could be made more concise, freeing space to (a) elaborate key proof ideas and initialization guidance, and (b) expand the experiments.

---

### Official Review · Reviewer_7Nk8 · 2025-10-26

**Soundness:** 2
**Presentation:** 2
**Contribution:** 2
**Rating:** 2
**Confidence:** 4

**Summary:**

This paper proposes a differentially private mechanism for server-assisted nonconvex optimization. The authors point out that existing approaches typically require gradient clipping to ensure differential privacy. To mitigate the effect of clipping-induced errors on convergence accuracy, they propose an approach that combines smoothed normalization and error compensation techniques, and achieves convergence to a neighborhood of a stable solution. In my opinion, the main contribution of this paper lies in removing the bounded gradient assumption commonly used in existing results.

My main concerns are as follows:

(i) Overstated and misleading statements (the algorithm is only applicable to server-assisted optimization (e.g., federated learning) rather than distributed optimization; see Weakness 1 for details);

(ii) Weak convergence results (the algorithm does not achieve accurate convergence, which has already been achieved in existing works; see Weaknesses 1 and 2 for deatils);

(iii) Lack of differential-privacy analysis (see Weakness 3 for deatils);

**Strengths:**

This paper removes the bounded gradient assumption used in existing differential-privacy results. The experimental evaluation is relatively thorough.

**Weaknesses:**

1. **Overstated and misleading statements:** In the Abstract, the authors state: "Surprisingly, there is no differentially private distributed method for smooth, non-convex optimization problems with convergence guarantees." This statement (made without qualification) is overly exaggerated, as several approaches have already been proposed (see, e.g., [r1], [r2], [r3]) for fully distributed nonconvex optimization or fully distributed bilevel nonconvex optimization. Moreover, these works have achieved both accurate convergence and differential privacy.

[r1] Chen J, Wang J, Zhang J F. Differentially private distributed nonconvex stochastic optimization with quantized communication. IEEE Transactions on Automatic Control, 2025 (arXiv version in 2024).

[r2] Chen Z, Wang Y. Locally differentially private decentralized stochastic bilevel optimization with guaranteed convergence accuracy. Forty-first International Conference on Machine Learning. 2024.

[r3] Yue X Y, Xiao J W, Liu X K, et al. Differentially private linearized ADMM algorithm for decentralized nonconvex optimization. IEEE Transactions on Information Forensics and Security, March, 2025.

In addition, the statements in the paper are quite misleading. For example, in the first paragraph of the Introduction and throughout the paper, terms such as "distributed optimization", "distributed setting", and "distributed gradient methods" are repeatedly used. From these descriptions, I was under the impression—up to page 6—that the authors were considering a fully distributed setting without any centralized server or aggregator. However, in Algorithm 1 on page 6, a centralized server is still required for computation and coordination, which contradicts the earlier claims as well as the title of the paper.

2. **Weak convergence results:** Theorem 1 only proves that Algorithm 1 (without considering privacy constraint) converges to a neighborhood $O(2R+\frac{L}{2}\gamma)$ of a stable point to problem (1), rather than achieving accurate convergence. Following Theorem 1, the authors explain that, by choosing $g_{i}^{0}$, e.g., $g_{i}^{0}=\nabla f_{i}(x^{0})+e$
with $e=(D/\sqrt{K+1},0,\ldots,0)$, they can ensure $R=\max_{i\in[1,n]}||\nabla f_{i}(x^{0})-g_{i}^{0}||=D(K+1)^{-\frac{1}{2}}$, and hence, obtain a convergence guarantee. However, it should be noted that $g_{i}^{0}$ is an initialization in Algorithm 1, which is typically a pre-selected constant. In this case, $K$ is also a pre-determined constant and cannot tend to infinity, thereby failing to ensure accurate convergence. Similar issues also arise in Theorem 2, which analyzes the convergence of Algorithm 1 under privacy constraints. What's worse, the optimization error established in Theorem 2 grows to infinity as the number of iterations $K$ tends to infinity (with an increasing rate of $O(K)$).
Therefore, the authors' claims of  "convergence guarantee" and "$\alpha$-NormEC achieves $(\varepsilon,\delta)$-DP and comes with convergence guarantees" are unconvincing.

3. **Lack of differential-privacy analysis:** Although the authors cite relevant literature (e.g., Abadi et al. (2016)), differential privacy (which is one of the main focuses of this paper) is not analyzed anywhere in the manuscript, including the Appendix. This omission makes the paper not self-contained.

**Questions:**

See the weaknesses above. In addition, I have the following questions:

1. In Algorithm 1, the authors also require the injected Gaussian noise to have bounded variance, which renders the statement on page 2 self-contradictory. (See the statement "While the method has been studied in the single-node setting, the convergence results rely on unrealistic and/or restrictive assumptions, such as symmetric gradient noise (Bu et al., 2024) and almost sure bounds on the gradient noise variance" on page 2).

2. Theorem 1 only proves convergence to a neighborhood of a stable solution to problem (1). Following Theorem 1, the authors state that "By proper choices of parameters", Algorithm 1 can achieve accurate convergence. However, it is unclear which parameters are being referred to and how they should be tuned?

3. Theorem 2 holds only when $\beta=\frac{1}{K+1}$ (see Corollary 2), which is important and should be explicitly clarified in the theorem statement.

4. In experimental setups (in Appendix on page 29), the authors simply state that "The train samples were randomly shuffled and distributed across 10 workers." However, random shuffling typically yields (near) IID data distribution. How is heterogeneity of the data distribution across workers/agents ensured?

5. Since Assumptions 1–2 already ensure the Lipschitz continuity of each objective function $f_{i}$, the latter part of Assumption 1–1) is redundant.

If the authors could avoid the inaccurate statements and address Weaknesses 2 and 3, I would consider raising the score.

---

### Official Review · Reviewer_DZsE · 2025-10-31

**Soundness:** 3
**Presentation:** 1
**Contribution:** 2
**Rating:** 2
**Confidence:** 4

**Summary:**

The paper proposes α-NormEC, a distributed optimization algorithm that integrates smoothed normalization with the EF21 error-feedback mechanism. The method addresses convergence challenges in non-convex, distributed, and differentially private (DP) optimization without assuming bounded gradients. Theoretical results show sublinear convergence in both non-private and private settings, including the provable utility bound for DP distributed optimization. Some experiments on CIFAR-10 with ResNet-20 and comparison between DP-SGD and DP-Clip21 are also included.

**Strengths:**

The paper presents an integration of smoothed normalization and EF21 that enables convergence without bounded-gradient assumptions. Theoretical analysis is sound and avoids restrictive conditions. The induction-based proof for state-dependent contractive operators is technically elegant. α-NormEC achieves provable convergence in the DP distributed setting.

**Weaknesses:**

1. There are already several existing works on generalizing clip21 into a DP version, for example "Double Momentum and Error Feedback for Clipping with Fast Rates and Differential Privacy". The proposed structures are almost identical to that shown in this paper except the final normalization step performed by the server. And the theoretical role of server normalization (SN) is not fully analyzed. While SN sometimes stabilizes training empirically, there is no clear theoretical justification or guidance on when it should be applied. From the provided the experiments, the normalization seems to compromise the performance.

2. There is a minor inconsistency in Assumption 1: if each local function fi is Li-smooth, then their average f is also L-smooth with L = (1/n)∑Li, so it is not necessary to restate the Lipschitz condition for f separately—only boundedness of f needs to be assumed.
Line 187-188, the sentence “Notice that the DP Gaussian noise variance (3) is scaled with the sensitivity Φ” is inaccurate. According to the standard DP-SGD formulation, the standard deviation, not the variance, scales linearly with Φ. Clarifying this would improve precision.

3. The paper also lacks comparison with SOTA DP-SGD benchmark on CIFAR10, e.g., "unlocking high-accuracy differentially private image classification through scale", "differentially private learning needs better features (or much more data)", "a theory to instruct differentially-private learning via clipping bias reduction". I cannot see clear privacy-utility tradeoff from the experiments. For example, I cannot find the test accuracy achieved by $(\epsilon=8, \delta=10^{-5})$.

4. In addition, the visualization quality could be improved. In several figures, the dashed and solid curves (e.g., Figure 2) are difficult to distinguish, which affects readability.

**Questions:**

1. There are already several existing works on generalizing clip21 into a DP version, for example "Double Momentum and Error Feedback for Clipping with Fast Rates and Differential Privacy". The proposed structures are almost identical to that shown in this paper except the final normalization step. Can the authors compare these results and explain how much the normalization can help, .e.g., under what precise conditions?

2. The convergence rate in Theorem 3 seems independent of α once normalization saturates. How should we intuitively understand this?

---

### Official Review · Reviewer_Dc5T · 2025-10-31

**Soundness:** 2
**Presentation:** 3
**Contribution:** 2
**Rating:** 2
**Confidence:** 4

**Summary:**

This paper uses smoothed normalization with an error compensation mechanism and proposes a new algorithm, DP–NormEC, for the differentially private setting. The authors aim to mitigate the negative effects of clipping in DP-SGD by replacing the clipping operation with a smoothed normalization step, which they claim provides better convergence behavior. They derive convergence guarantees for both the non-private and private versions of the algorithm and provide empirical results on CIFAR-10 to support their claims.

**Strengths:**

- The topic is relevant, as gradient clipping remains a key challenge for differential privacy in optimization and federated learning.
- The idea of combining smoothed normalization with error compensation is reasonable, and the theoretical analysis is presented clearly.

**Weaknesses:**

- The novelty is limited. In the non-private case, DP–NormEC appears to follow from EC21 once smooth normalization is assumed contractive, so I am unsure that it matches the level required for ICLR paper.
- The paper lacks a formal privacy analysis, despite presenting itself as a differentially private algorithm. The seven in appendix only provide big O guarantees which is not enough for the different privacy results
- The federated learning setting is not convincingly addressed: only one-step updates per agent are considered, which limits both the technical contribution and the generality of the approach.
- Both the clipping threshold ($\tau$) and smoothing parameter ($\alpha$) are sensitive. Figures 3 and 15 show that performance depends strongly on $\alpha$, yet the paper suggests that tuning normalization is easier than clipping without clear justification, whereas it is a key point to motivate the method
- The experimental validation is weak: results are limited to CIFAR-10, seemingly from a single run, with no error bars or statistical analysis.
- The server normalization step can degrade utility when not needed, which is not discussed.

**Questions:**

- Can you compare your work to *"Double Momentum and Error Feedback for Clipping with Fast Rates and Differential Privacy", Rustem Islamov, Samuel Horvath, Aurelien Lucchi, Peter Richtarik, Eduard Gorbunov*
- Can you provide explicit privacy guarantees and clarify how the noise is calibrated?
- Why is tuning $\alpha$ expected to be easier than tuning $\tau$, given its sensitivity in experiments?
- How would the approach extend to more realistic federated settings with multiple local updates or partial participation?

---

### Meta-Review · Area_Chair_x3zz · 2025-12-10

**Summary:**

No response was given to the weaknesses highlighted by the reviewers. I therefore consider that the authors agree with these criticisms and that they cannot be easily addressed.

**Reviewer Scores:**

N/A

---

### Decision · Program_Chairs · 2026-01-26

Reject